# Attention-based Neural Cellular Automata

**Mattie Tesfaldet**
McGill University, Mila

**Derek Nowrouzezahrai**
McGill University, Mila

**Christopher Pal**[*]
Polytechnique Montréal, Mila

## Abstract

Recent extensions of Cellular Automata (CA) have incorporated key ideas from modern deep learning, dramatically extending their capabilities and catalyzing a new family of Neural Cellular Automata (NCA) techniques. Inspired by Transformer-based architectures, our work presents a new class of *attention-based* NCAs formed using a spatially localized—yet globally organized—self-attention scheme. We introduce an instance of this class named *Vision Transformer Cellular Automata (ViTCA)*. We present quantitative and qualitative results on denoising autoencoding across six benchmark datasets, comparing ViTCA to a U-Net, a U-Net-based CA baseline (UNetCA), and a Vision Transformer (ViT). When comparing across architectures configured to similar parameter complexity, ViTCA architectures yield superior performance across all benchmarks and for nearly every evaluation metric. We present an ablation study on various architectural configurations of ViTCA, an analysis of its effect on cell states, and an investigation on its inductive biases. Finally, we examine its learned representations via linear probes on its converged cell state hidden representations, yielding, on average, superior results when compared to our U-Net, ViT, and UNetCA baselines.

## 1 Introduction

Recent developments at the intersection of two foundational ideas—Artificial Neural Networks (ANNs) and Cellular Automata (CA)—have led to new approaches for constructing Neural Cellular Automata (NCA). These advances have integrated ideas such as variational inference [7], U-Nets [26], and Graph Neural Networks (GNNs) [15] with promising results on problems ranging from image synthesis [7, 20, 21] to Reinforcement Learning (RL) [6, 22]. Transformers are another significant development in deep learning [41], but, until now, have not been examined under an NCA setting.

Vision Transformers (ViTs) [13] have emerged as a competitive alternative to Convolutional Neural Network (CNN) [56] architectures for computer vision, such as Residual Networks (ResNets) [45]. ViTs leverage the self-attention

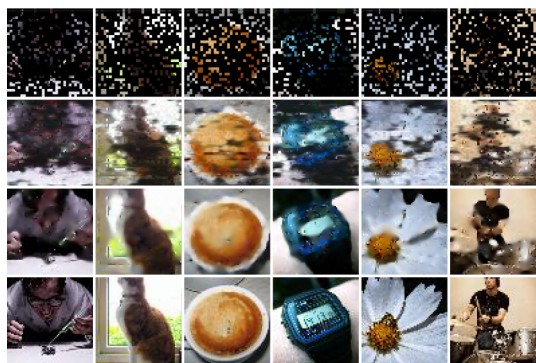

Figure 1: **ViT** *vs*. **ViTCA** for denoising Tiny ImageNet [49] validation set images with $2{\times}2$ pixel masks covering $75\%$ of the image. *Top-to-bottom*: noisy input, ViT, ViTCA, and ground truth.

mechanisms of original Transformers [41], which have emerged as the dominant approach for sequence modelling in recent years. Our work combines foundational ideas from Transformers and ViTs, leading to a new class of NCAs: **Vision Transformer Cellular Automata (ViTCA)**.

---

[*]Canada CIFAR AI Chair

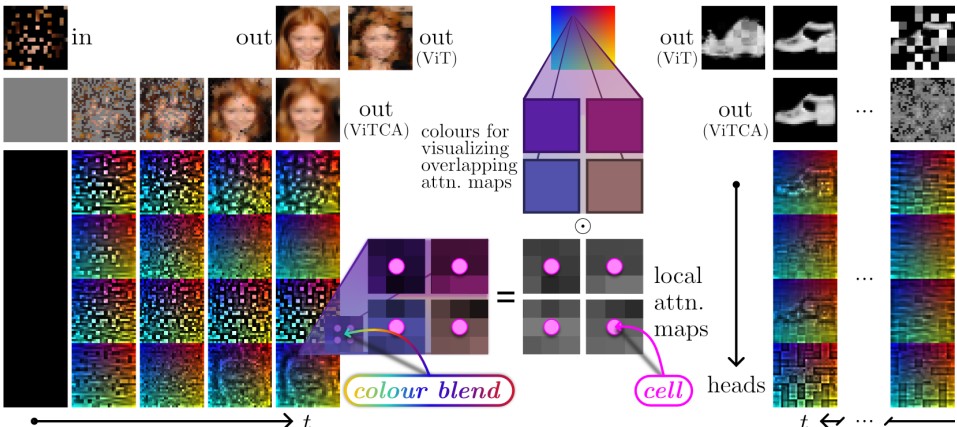

Figure 2: **Global self-organization manifested within localized self-attention**. Despite operating in spatially local neighbourhoods about a cell, over time the localized (multi-head) self-attention in ViTCA experiences a *global* self-organization admitted by its NCA nature. This circumvents the quadratic complexity of explicit global self-attention (w.r.t. input size) with a linear amortization over time (recurrent CA iterations), enabling effective per-pixel dense processing. *Middle*: visualizing local attention maps about each cell as colour-coded "splats" blended together in overlapping regions, producing a "splat map" [58]. *Left, right*: ViTCA iterations on a cell grid, updated from a seed state to a converged state, given a noisy input image to denoise. For each head of the cells' local attention maps, there is global agreement on the types of features to attend to (*e.g.*, foreground contours, noise, background). Enveloping ViT by the NCA paradigm dramatically improves its output fidelity.

An effective and ubiquitous Transformer-based learning technique for Natural Language Processing (NLP) pre-training is the unsupervised task of Masked Language Modelling (MLM), popularized by the BERT language model [34]. The success of MLM-based techniques has similarly inspired recent work re-examining the classical formulation of Denoising Autoencoders (DAEs) [51], but for ViTs [3, 13, 28], introducing tasks such as Masked Image Encoding [16] and Masked Feature Prediction [24] for image and video modelling, respectively. This simple yet highly-scalable strategy of masked-based unsupervised pre-training has yielded promising transfer learning results on vision-based downstream tasks such as object detection and segmentation, image classification, and action detection, even outperforming supervised pre-training [16, 24]. We examine training methodologies for ViTCA within a DAE setting and perform extensive controlled experiments benchmarking these formulations against modern state of the art architectures, with favourable outcomes, *e.g.*, Fig. 1.

Our contributions are as follows: *first*—to the best of our knowledge—our work is the first to extend NCA methodologies with key Transformer mechanisms, *i.e.*, self-attention and positional encoding (and embedding), with the beneficial side-effect of circumventing the quadratic complexity of self-attention; *second*, our ViTCA formulation allows for lower model complexity (by limiting ViT depth) while retaining expressivity through CA iterations on a controlled state—all with the same encoder weights. This yields a demonstrably more parameter-efficient [20] ViT-based model. Importantly, ViTCA mitigates the problems associated with the explicit tuning of ViT depth originally needed to improve performance (*i.e.*, we use a depth of 1). With ViTCA, we simply iterate until cell state convergence. Since ViT (and by extension, ViTCA) employs Layer Normalization (LN) [43] at each stage of its processing, it is a fairly contractive model capable of fixed-point convergence guarantees [32].

In relation to our first contribution, ViTCA respects CA requirements, most importantly that computations remain localized about a cell and its neighbourhood. As such, we modify the global self-attention mechanisms of a ViT to respect this locality requirement (Fig. 2). Localized self-attention is not a new idea [4, 12, 19, 27]; however, because cells contain state information that depends on its previous state, over CA iterations the effective receptive field of ViTCA's localized self-attention grows increasingly larger until eventually incorporating information implicitly across all cells. Thus, admitting global propagation of information from spatially localized self-attention. Moreover, due to the self-organizing nature of NCAs, self-organization also manifests itself within the localized self-attention, resulting in a globally agreed-upon arrangement of local self-attention. Thus, circumventing the quadratic complexity of explicit global self-attention (w.r.t. the input size)

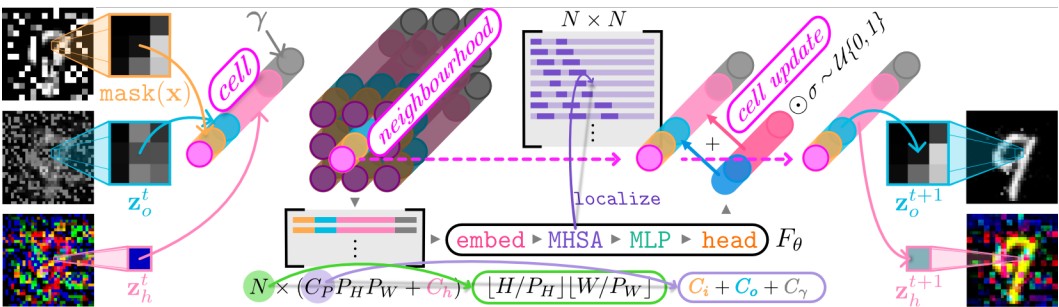

Figure 3: **Computational overview**. NCAs use a stateful lattice of *cells*, each storing information along channels, to promote desired behaviour over the course of an evolutionary cycle. Starting from an initial seed, each cell state evolves at discrete time steps according to a homogeneous, learned *update rule* applied either synchronously or asynchronously ($\sigma$). This update depends on the current cell state and that of its *neighbours* (pictured is the *Moore neighbourhood* [1]). In **ViTCA**, each cell is represented as a vector where the first $C_i P_H P_W$ channels contain a $P_H \times P_W$ noisy input image patch (mask($\mathbf{x}$)), the next $C_o P_H P_W$ channels contain the current output patch ($\mathbf{z}_o^t$), the following $C_h$ channels contain undefined data hidden from the loss that can be used to encode additional information ($\mathbf{z}_h^t$), and (optionally) the remaining $C_\gamma P_H P_W$ channels contain positional information ($\boldsymbol{\gamma}$). The update rule ($F_\theta$) is a modified ViT [13] whose self-attention mechanism is locally constrained to each cell's neighbourhood (localize).

through a linear amortization over time, and increasing the feasibility of per-pixel dense processing (as we demonstrate). This globally consistent and complex behaviour, which arises from strictly local interactions, is a unique feature of NCAs and confers performance benefits which we observe both qualitatively and quantitatively when comparing ViT and ViTCA for denoising autoencoding.

## 2 Background and related work

**Neural Cellular Automata.** Cellular Automata are algorithmic processes motivated by the biological behaviours of cellular growth and, as such, are capable of producing complex emergent (global) dynamics from the iterative application of comparatively simple (localized) rules [60]. *Neural* Cellular Automata present a more general CA formulation, where the evolving cell states are represented as (typically low-dimensional) vectors and the update rule dictating their evolution is a differentiable function whose parameters are learned through backpropagation from a loss, rather than a handcrafted set of rules [30, 35, 59]. Neural net-based formulations of CAs in the NeurIPS community can be traced back to the early work of [59], where only small and simple models were examined. Recent formulations of NCAs have shown that when leveraging the power of deep learning techniques enabled by advances in hardware capabilities—namely highly-parallelizable differentiable operations implemented on GPUs—NCAs can be tuned to learn surprisingly complex desired behaviour, such as semantic segmentation [31]; common RL tasks such as cart-pole balancing [22], 3D locomotion [6], and Atari game playing [6]; and image synthesis [7, 20, 21]. Although these recent formulations rely on familiar compositions of convolutions and non-linear functions, it is important to highlight that NCAs are fundamentally not equivalent to "very-deep" CNNs (*vs.* [35]), or any other feedforward architecture (*e.g.*, ResNets [45]), particularly, in the same way that a Recurrent Neural Network (RNN) is not equivalent: CNNs and other feedforward architectures induce a directed *acyclic* computation graph (*i.e.*, a finite impulse response), whereas NCAs (and RNNs) induce a directed *cyclic* computation graph (*i.e.*, an infinite impulse response), where stateful data can additionally be manipulated using (learned) feedback loops and/or time-delayed controls. As such, NCAs can be viewed as a type of RNN, and both (N)CAs and RNNs are known to be Turing complete [11, 54, 57, 59].[2]

**Vision Transformers.** Vision Transformers [13] are an adaptation of Transformers [41] to vision-based tasks like image classification. In contrast to networks built from convolutional layers, ViTs rely on *self-attention* mechanisms operating on tokenized inputs. Specifically, input images are divided into non-overlapping patches, then fed to a Transformer after undergoing a linear patch projection with an embedding matrix. While ViTs provide competitive image classification performance, the

---

[2]In the case of (N)CAs, a Turing complete example is the *Rule 110* elementary CA [11, 54]

quadratic computational scaling of global self-attention limits their applicability in high-dimensional domains, *e.g.*, per-pixel dense processing. Recent developments have attempted to alleviate such efficiency limitations [9, 10, 14, 17], one notable example being Perceiver IO [5, 8] with its use of cross-attention. We refer interested readers to a comprehensive survey on ViTs [18].

## 3  Vision Transformer Cellular Automata (ViTCA)

Building upon NCAs and ViTs, we propose a new class of *attention-based* NCAs formed using a spatially localized—yet globally organized—self-attention scheme. We detail an instance of this class, ViTCA, by first reviewing its backbone ViT architecture before describing the "pool sampling"-based training process for the ViTCA update rule (see overview in Fig. 3).

**Input tokenization.**  ViT starts by dividing a $C_i \times H \times W$ input image $\mathbf{X}$ into $N$ non-overlapping $P_H \times P_W$ patches ($16 \times 16$ in the original work [13]), followed by a linear projection of the flattened image patches with an embedding matrix $\mathbf{E} \in \mathbb{R}^{L \times d}$ (Fig. 3 embed), where $L = C_i P_H P_W$, to produce initial tokens $\mathbf{T}' \in \mathbb{R}^{N \times d}$. Next, a handcrafted positional encoding [41] or learned positional embedding $\boldsymbol{\gamma} \in \mathbb{R}^{N \times d}$ [13] is added to tokens to encode positional information and break permutation invariance. Finally, a learnable class token is appended to the token sequence, resulting with $\mathbf{T} \in \mathbb{R}^{(N+1) \times d}$. For the purposes of our task, we omit this token in all ViT-based models. In ViTCA, the input to the embedding is a flattened cell grid $\mathbf{Z} \in \mathbb{R}^{N \times L}$ where $L = C_P P_H P_W + C_h$, $C_P = C_i + C_o + C_\gamma$, $C_h$ is the cell hidden size, $C_o$ is the number of output image channels (one or three for grayscale or RGB), and $C_\gamma$ is the positional encoding size when positional encoding is (optionally) concatenated to each cell rather than added to the tokens [29].

**Multi-head self-attention (MHSA).**  Given a sequence of tokens $\mathbf{T}$, self-attention estimates the relevance of one token to all others (*e.g.*, which image patches are likely to appear together in an image) and aggregates this global information to update each token. This encodes each token in terms of global contextual information, and does so using three learned weight matrices: $\mathbf{W}_Q \in \mathbb{R}^{d \times d}$, $\mathbf{W}_K \in \mathbb{R}^{d \times d}$, and $\mathbf{W}_V \in \mathbb{R}^{d \times d}$. $\mathbf{T}$ is projected onto these weight matrices to obtain Queries $\mathbf{Q} = \mathbf{T}\mathbf{W}_Q$, Keys $\mathbf{K} = \mathbf{T}\mathbf{W}_K$, and Values $\mathbf{V} = \mathbf{T}\mathbf{W}_V$. The self-attention layer output $\mathtt{SA} \in \mathbb{R}^{N \times d}$ is:

$$\mathtt{SA} = \mathtt{softmax}\left(\mathbf{Q}\mathbf{K}^T / \sqrt{d}\right) \mathbf{V} . \tag{1}$$

*Multi-head* self-attention employs many sets of weight matrices, $\{\mathbf{W}_{Q_i}, \mathbf{W}_{K_i}, \mathbf{W}_{V_i} \in \mathbb{R}^{d \times (d/h)} \mid i = 0, ..., (h-1)\}$. The outputs of $h$ self-attention *heads* are concatenated into $(\mathtt{SA}_0, ..., \mathtt{SA}_{h-1}) \in \mathbb{R}^{N \times d}$ and projected onto a weight matrix $\mathbf{W} \in \mathbb{R}^{d \times d}$ to produce $\mathtt{MHSA} \in \mathbb{R}^{N \times d}$. Self-attention explicitly models global interactions and is more flexible than grid-based operators (*e.g.*, convolutions) [33, 38], but its quadratic cost in time and memory limits its applicability to high resolution images.

**Spatially localizing self-attention.**  The global nature of self-attention directly conflicts with the spatial locality constraint of CAs; in response, we limit the connectivity structure of the attention operation to each cell's neighbourhood. This can be accomplished by either masking each head's attention matrix ($\mathbf{A} = \mathtt{softmax}(\cdots) \in \mathbb{R}^{N \times N}$ in Eq. 1) with a banded matrix representing local connectivity (*e.g.*, Fig. 3 localize), or more efficiently,

$$\mathbf{A}^\star = \mathtt{softmax}\left(\mathbf{A}' / \sqrt{d}\right) \quad \text{s.t.} \ (\mathbf{A}')_{ij} = \sum_l (\mathbf{Q})_{il}(\mathbf{K})_{jl} \quad (2) \qquad \Big| \qquad \mathtt{SA}^\star = \mathbf{A}^\star \mathbf{V} \quad (3)$$

with $(\mathbf{V})_{jl}$ where $i = \{0, ..., (N-1)\}$, $j = \{(i+n_w+n_h), ..., i, ..., (i-n_w-n_h)\}$, and $l = \{0, ..., (d-1)\}$, and with $n_w = \{-\lfloor N_W/2 \rfloor, ..., 0, ..., \lfloor N_W/2 \rfloor\}$ and $n_h = \{-W\lfloor N_H/2 \rfloor, ..., 0, ..., W\lfloor N_H/2 \rfloor\}$. Here, we assume top-left-to-bottom-right input flattening. Instead of explicitly computing the global self-attention matrix $\mathbf{A} \in \mathbb{R}^{N \times N}$ then masking it, this approach circumvents the $\mathcal{O}(N^2 d)$ computation in favour of an $\mathcal{O}(NMd)$ alternative that indexes the necessary rows and columns *during* self-attention. The result is a localized self-attention matrix $\mathbf{A}^\star \in \mathbb{R}^{N \times M}$, where $M = N_H N_W \ll N$. As we show in our experiments, ViTCA is still capable of global self-attention despite its localization, by leveraging stored state information across cells and their global self-organization during CA iterations (Fig. 2).

Following $\mathtt{MHSA}$ is a multilayer perceptron (Fig. 3 MLP) with two layers and a GELU non-linearity. We apply Layer Normalization (LN) [43] before $\mathtt{MHSA}$ and $\mathtt{MLP}$, and residual connections afterwards, forming a single encoding block. We use an MLP head (Fig. 3 head) to decode to a desired output,

with LN applied to its input, finalizing the ViTCA update rule $F_\theta$. In our experiments, ViT's `head` decodes directly into an image output whereas ViTCA decodes into update vectors added to cells.

## 3.1 Update rule training procedure

To train the ViTCA update rule, we follow a "pool sampling"-based training process [7, 30] along with a curriculum-based masking/noise schedule when corrupting inputs. During odd training iterations, we uniformly initialize a minibatch of cells $\mathbf{Z} = (\mathbf{Z}_1, ..., \mathbf{Z}_b)$ with constant values (0.5 for output channels, 0 for hidden—see Appendix A.2 for alternatives), then inject the masked input `mask(X)` (see Sec. 4.1). After input injection, we asynchronously update cells ($\sigma = 50\%$ update rate) using $F_\theta$ for $T \sim \mathcal{U}\{8, 32\}$ recurrent iterations. We retrieve output $\mathbf{Z}_o$ from the cell grid and apply an $L_1$ loss against the ground truth $\mathbf{X}$. We also apply overflow losses to penalize cell output values outside of [0,1] and cell hidden values outside of [-1,1]. We use $L_2$ normalization on the gradient of each parameter in $\theta$. After backpropagation, we append the updated cells and their ground truths to a pool $\mathcal{P}$ which we then shuffle and truncate up to the first $N_\mathcal{P}$ elements. During even training iterations, we retrieve a minibatch of cells and their ground truths from $\mathcal{P}$ and process them as above. This encourages $F_\theta$ to guide cells towards a stable fixed-point. Alg. 1 in Appendix A details this process.

# 4 Experiments

Here we examine ViTCA through extensive experiments. We begin with experiments for denoising autoencoding, then an ablation study followed by various qualitative analyses, before concluding with linear probing experiments on the learned representations for MNIST [50], FashionMNIST [42], and CIFAR10 [53]. We provide an extension to our experiments in Appendix A.

**Baseline models and variants.** Since we are performing pixel level reconstructions, we create a ViT baseline in which the class token has been removed. This applies identically for ViTCA. Unless otherwise stated, for our ViT and ViTCA models we use a patch size of $1 \times 1$ ($P_H = P_W = 1$), and only a single encoding block with $h = 4$ `MHSA` heads, `embed` size $d = 128$, and `MLP` size of 128. For ViTCA, we choose $N_H = 3$ and $N_W = 3$ (*i.e.*, the *Moore neighbourhood* [1]). We also compare with a U-Net baseline similar to the original formulation [48], but based on the specific architecture from [37]. Since most of our datasets consist of $32 \times 32$ (resampled) images, we only have two downsampling steps as opposed to five. We implement a U-Net-based CA (UNetCA) baseline consisting of a modified version of our U-Net with 48 initial output feature maps as opposed to 24 and with all convolutions except the first changed to $1 \times 1$ to respect typical NCA restrictions [7, 30].

## 4.1 Denoising autoencoding

We compare between our baseline models and a number of ViTCA variants in the context of denoising autoencoding. We present test set results across six benchmark datasets: a land cover classification dataset intended for representation learning (LandCoverRep) [25], MNIST, CelebA [47], FashionMNIST, CIFAR10, and Tiny ImageNet (a subset of ImageNet [49]). All datasets consist of $32 \times 32$ resampled images except Tiny ImageNet, which is at $64 \times 64$ resolution. During testing, we use all masking combinations, chosen in a fixed order, and we update cells using a fixed number of iterations ($T = 64$). See Tab. 1 for quantitative results.

Briefly mentioned in Sec. 3.1, we employ a masking strategy inspired by Curriculum Learning (CL) [23, 52] to ease training. This schedule follows a geometric progression of difficulty—tied to training iterations—maxing out at 10K training iterations. Specifically, masking starts at covering 25% of the input with $1 \times 1$ patches of noise (dropout for RGB inputs, Gaussian for grayscale), then at each shift in difficulty, new masking configurations are added to the list of available masking configurations in the following order: $(2^0 \times 2^0, 50\%), (2^0 \times 2^0, 75\%), (2^1 \times 2^1, 25\%), (2^1 \times 2^1, 50\%), (2^1 \times 2^1, 75\%), ..., (2^2 \times 2^2, 75\%)$. Masking configurations are randomly chosen from this list.

We initialize weights/parameters using He initialization [46], except for the final layer of CA-based models, which are initialized to zero [30]. Unless otherwise stated, we train for $I = 100$K iterations, use a minibatch size $b = 32$, AdamW optimizer [36], learning rate $\eta = 10^{-3}$ with a cosine annealing schedule [40], pool size $N_\mathcal{P} = 1024$, and cell hidden channel size $C_h = 32$. In the case of Tiny ImageNet, $b = 8$ to accommodate training on a single GPU (48GB Quadro RTX 8000). Training typically lasts a day at most, depending on the model. Due to the recurrent iterations required per training step, CA-based models take the longest to train. To alleviate memory limitations for some of our experiments, we use gradient checkpointing [44] during CA iterations at the cost of

| | **LandCoverRep** | | | | **CelebA** | | | | **MNIST** | | | |
|---|---|---|---|---|---|---|---|---|---|---|---|---|
| | PSNR↑ | SSIM↑ | LPIPS↓ | #Params. | PSNR↑ | SSIM↑ | LPIPS↓ | #Params. | PSNR↑ | SSIM↑ | LPIPS↓ | #Params. |
| **Baselines** U-Net | 33.94 | 0.934 | **0.099** | 106.6K | 26.23 | 0.906 | 0.075 | 106.6K | 23.43 | 0.897 | 0.049 | 104.5K |
| ViT | 30.64 | 0.893 | 0.135 | 83.9K | 19.70 | 0.779 | 0.237 | 83.9K | 16.02 | 0.631 | 0.254 | 83.4K |
| UNetCA | 33.94 | _0.935_ | _0.102_ | 54.0K | 25.66 | 0.882 | 0.091 | 54.0K | 25.61 | 0.929 | 0.034 | 52.0K |
| ViTCA | 33.80 | 0.932 | _0.102_ | 92.5K | 26.53 | _0.913_ | _0.066_ | 92.5K | _27.01_ | 0.940 | _0.028_ | 91.7K |
| **Variants** ViTCA-32 | _34.00_ | _0.935_ | 0.103 | 92.5K | **27.01** | **0.920** | **0.060** | 92.5K | **27.68** | **0.946** | **0.026** | 91.7K |
| ViTCA-32xy | **34.06** | **0.936** | 0.106 | 92.8K | _26.75_ | 0.898 | 0.072 | 92.8K | 26.97 | _0.942_ | _0.028_ | 92.0K |
| ViTCA-i | 33.49 | 0.929 | 0.108 | 54.7K | 26.10 | 0.904 | 0.074 | 54.7K | 26.03 | 0.930 | 0.033 | 54.3K |
| ViTCA-i16 | 33.74 | 0.932 | 0.106 | 54.7K | 26.61 | 0.912 | _0.066_ | 54.7K | 26.42 | 0.935 | 0.031 | 54.3K |
| ViTCA-ixy | 33.75 | 0.933 | 0.107 | 54.8K | 26.51 | 0.894 | 0.076 | 54.8K | 25.95 | 0.933 | 0.033 | 54.4K |
| ViTCA-i16xy | 33.93 | _0.935_ | 0.108 | 54.8K | 26.68 | 0.898 | 0.074 | 54.8K | 26.28 | 0.936 | 0.031 | 54.4K |
| | **FashionMNIST** | | | | **CIFAR10** | | | | **Tiny ImageNet** | | | |
| **Baselines** U-Net | 24.19 | 0.852 | 0.126 | 104.5K | 25.62 | 0.855 | 0.131 | 106.6K | 21.93 | 0.775 | 0.203 | 106.6K |
| ViT | 16.28 | 0.519 | 0.397 | 83.4K | 20.99 | 0.744 | 0.237 | 83.9K | 17.80 | 0.598 | 0.355 | 83.9K |
| UNetCA | 23.67 | 0.854 | 0.123 | 52.0K | 25.49 | 0.851 | 0.129 | 54.0K | 21.78 | 0.773 | 0.204 | 54.0K |
| ViTCA | 23.80 | 0.855 | 0.117 | 91.7K | 25.61 | 0.856 | 0.127 | 92.5K | 21.58 | 0.772 | 0.215 | 92.5K |
| **Variants** ViTCA-32 | **24.91** | **0.874** | **0.098** | 91.7K | _26.05_ | _0.864_ | _0.122_ | 92.5K | 21.94 | 0.781 | 0.202 | 92.5K |
| ViTCA-32xy | _24.55_ | _0.869_ | _0.102_ | 92.0K | **26.14** | **0.866** | **0.120** | 92.8K | **22.03** | **0.783** | **0.199** | 92.8K |
| ViTCA-i | 22.84 | 0.827 | 0.139 | 54.3K | 25.42 | 0.853 | 0.132 | 54.7K | 21.75 | 0.776 | 0.211 | 54.7K |
| ViTCA-i16 | 23.32 | 0.839 | 0.127 | 54.3K | 25.65 | 0.856 | 0.128 | 54.7K | 21.72 | 0.774 | 0.213 | 54.7K |
| ViTCA-ixy | 23.54 | 0.848 | 0.123 | 54.4K | 25.85 | 0.861 | 0.125 | 54.8K | 21.95 | _0.782_ | _0.201_ | 54.8K |
| ViTCA-i16xy | 23.59 | 0.848 | 0.121 | 54.4K | 25.98 | 0.863 | 0.123 | 54.8K | 21.99 | _0.782_ | _0.201_ | 54.8K |

Table 1: Comparing denoising autoencoding results between baselines and ViTCA variants. ViTCA variants include: 32 (32 heads), 16 (16 heads), i (inverted bottleneck), xy (xy-coordinate positional encoding). Boldface and underlined values denote the best and second best results. Metrics include Peak Signal-to-Noise Ratio (PSNR; dB), Structural Similarity Index Measure (SSIM; values in $[0, 1]$) [55], Learned Perceptual Image Patch Similarity (LPIPS; values in $[0, 1]$) [39].

backpropagation duration and slight variations in gradients due to its effect on round-off propagation. We also experiment with a cell fusion and mitosis scheme as an alternative. See Appendix A for details on runtime performance, gradient checkpointing, and fusion and mitosis.

Amongst baselines, ViTCA outperforms on most metrics across the majority of datasets used (10 out of 18). Exceptions include LandCoverRep, where UNetCA universally outperforms by a small margin, likely due to the texture-dominant imagery being amenable to convolutions. Notably, ViTCA strongly outperforms on MNIST. Although MNIST is a trivial dataset for common tasks such as classification, our masking/noise strategy turns it into a challenging dataset for denoising autoencoding, *e.g.*, it is difficult for even a human to classify a $32 \times 32$ MNIST digit 75% corrupted by $4 \times 4$ patches of Gaussian noise. We hypothesize that when compared to convolutional models, ViTCA's weaker inductive biases (owed to attention [5, 8]) immediately outperform these models when there are large regions lacking useful features, *e.g.*, MNIST digits cover a small space in the canvas. This is not the case with FashionMNIST, where the content is more filled out. Between baselines and ViTCA variants, ViTCA-32 (32 heads) and 32xy (xy-coordinate positional encoding) outperform all models by large margins, demonstrating the benefits of multi-head self-attention. We also experiment with a parameter-reduced (by ∼60%), inverted bottleneck variant where $d = 64$ and MLP size is 256, often with a minimal reduction in performance.

### 4.1.1 Ablation study

In Tab. 2 we perform an ablation study using the baseline ViTCA model above as reference on CelebA. Results are ordered in row-wise blocks, top-to-bottom. Specifically, we examine the impact of varying the cell hidden size $C_h$; the embed size $d$; the number of MHSA heads $h$; the depth (# encoders), comparing both ViTCA (used throughout the table) with ViT; and in the last block we examine the impact of various methods of incorporating positional information into the model.

Specifically, we examine the use of: (1) a xy-coordinate-based positional encoding *concatenated* ("injected") to cells, and; (2) a Transformer-based positional encoding (or embedding, if learned) *added* into `embed`. These two categories are subdivided into: (1a) sincos5—consisting of handcrafted Fourier features [29] with four doublings of a base frequency, *i.e.*, $\gamma = (\sin 2^0 \pi p, \cos 2^0 \pi p, ..., \sin 2^{J-1}\pi p, \cos 2^{J-1}\pi p) \in \mathbb{R}^{N \times (4JP_H P_W)}$ where $J = 5$ and $p$ is the pixel coordinate (normalized to [-1,1]) for each pixel the cell is situated on (one pixel since $P_H = P_W = 1$); (1b) sincos5xy—consisting of both Fourier features and explicit xy-coordinates concatenated; (1c) xy—only xy-coordinates; (2a) handcrafted (our baseline approach)—sinusoidal encoding $\gamma \in \mathbb{R}^{N \times d}$ similar to (1a) but following a Transformer-based approach [41], and; (2b) learned—learned embedding $\gamma \in \mathbb{R}^{N \times d}$ following the original ViT approach [13]. To further test the self-organizing capabilities of ViTCA, we also include: (3) none—no explicit positioning provided, where we let the cells localize themselves.

As shown in Tab. 2, ViTCA benefits from an increase to most CA and Transformer-centric parameters, at the cost of computational complexity and/or an increase in parameter count. A noticeable decrease in performance is observed when `embed` size $d = 512$, most likely due to the vast increase in parameter count necessitating more training. In the original ViT, multiple encoding blocks were needed before the model could exhibit performance equivalent to their baseline CNN [13], as verified in our ablation with our ViT. However, for ViTCA we notice an inverse relationship of the effect of Transformer depth, causing a divergence in cell state. It is not clear why this is the case, as we have observed that the LN layers and overflow losses otherwise encourage a contractive $F_\theta$. This is an investigation we leave for future work. Despite the benefits of increasing $h$, we use $h = 4$ for our baseline to optimize runtime performance. Finally, we show that ViTCA does not dramatically suffer when no explicit positioning is used—in contrast to typical Transformer-based models—as cells are still able to localize themselves by relying on their stored hidden information.

|  |  | PSNR ↑ | SSIM ↑ | LPIPS ↓ | # Params. |
|---|---|---|---|---|---|
| **Hidden dim** | 8 | 25.61 | 0.898 | 0.086 | 86.3K |
| | 16 | 26.11 | 0.909 | 0.070 | 88.4K |
| | *32* | 26.53 | 0.913 | 0.066 | 92.5K |
| | 64 | 26.53 | 0.913 | 0.066 | 100.7K |
| | 128 | 26.51 | 0.912 | 0.066 | 117.2K |
| | 256 | 26.77 | 0.915 | **0.063** | 150.1K |
| | 512 | **26.78** | **0.916** | **0.063** | 215.9K |
| **Embed dim** | 8[†] | 21.67 | 0.814 | 0.258 | 2.0K |
| | 16[†] | 23.22 | 0.853 | 0.183 | 4.5K |
| | 32[†] | 24.94 | 0.875 | 0.110 | 10.9K |
| | 64[†] | 25.69 | 0.898 | 0.084 | 29.9K |
| | *128[†]* | 26.05 | 0.904 | 0.075 | 92.5K |
| | 256[†] | **26.36** | **0.911** | **0.067** | 316.0K |
| | 512[†] | 19.93 | 0.768 | 0.274 | 1.2M |
| **Heads** | 1 | 25.01 | 0.890 | 0.096 | 76.0K |
| | *4* | 26.53 | 0.913 | 0.066 | 92.5K |
| | 8 | 26.77 | 0.916 | 0.062 | 92.5K |
| | 16 | 26.78 | 0.917 | 0.062 | 92.5K |
| | 32 | **27.01** | **0.920** | **0.060** | 92.5K |
| | 64 | 26.94 | 0.919 | 0.061 | 92.5K |
| **Depth** | *ViTCA–1* | **26.53** | **0.913** | **0.066** | 92.5K |
| | ViTCA–2[†] | 10.82 | 0.225 | 0.771 | 175.3K |
| | ViTCA–3[†] | 9.70 | 0.165 | 0.793 | 258.0K |
| | *ViT–1* | 19.70 | 0.779 | 0.237 | 83.9K |
| | ViT–2[†] | 25.20 | 0.900 | 0.074 | 166.7K |
| | ViT–3[†] | **26.10** | **0.914** | **0.065** | 249.4K |
| **PE type** | sincos5 | 26.92 | 0.917 | 0.062 | 95.1K |
| | sincos5xy | **27.00** | **0.919** | **0.059** | 95.3K |
| | xy | 26.45 | 0.894 | 0.077 | 92.8K |
| | *handcrafted* | 26.53 | 0.913 | 0.066 | 92.5K |
| | learned | 26.16 | 0.910 | 0.071 | 223.6K |
| | none | 26.28 | 0.890 | 0.081 | 92.5K |

Table 2: Quantitative ablation for denoising autoencoding with ViTCA (unless otherwise stated via prefix) on CelebA [47]. Boldface and underlining denote best and second best results. Italicized items denote baseline configuration settings. [†]Trained with gradient checkpointing [44], which slightly alters round-off error during backpropagation, resulting in slight variations of results compared to training without checkpointing. See Appendix A.2.

### 4.1.2 Cell state analysis

Here we provide an empirically-based qualitative analysis on the effects ViTCA and UNetCA have on cell states through several experiments with our pre-trained models (Fig. 4 (a,b,c)). We notice that in general, ViTCA indefinitely maintains cell state stability while UNetCA typically induces a divergence past a certain point. An extended analysis is available in Appendix A.3.

**Damage resilience.** Shown in Fig. 4 (a), we damage a random $H/2 \times W/2$ patch of cells with random values $\sim \mathcal{U}(-1,1)$ twice in succession. ViTCA is able to maintain cell stability despite not being trained to deal with such noise, while UNetCA induces a divergence. Note both models are simultaneously performing the typical denoising task. We also note that ViTCA's inherent damage resilience is in contrast to recent NCA formulations that required explicit training for it [7, 30].

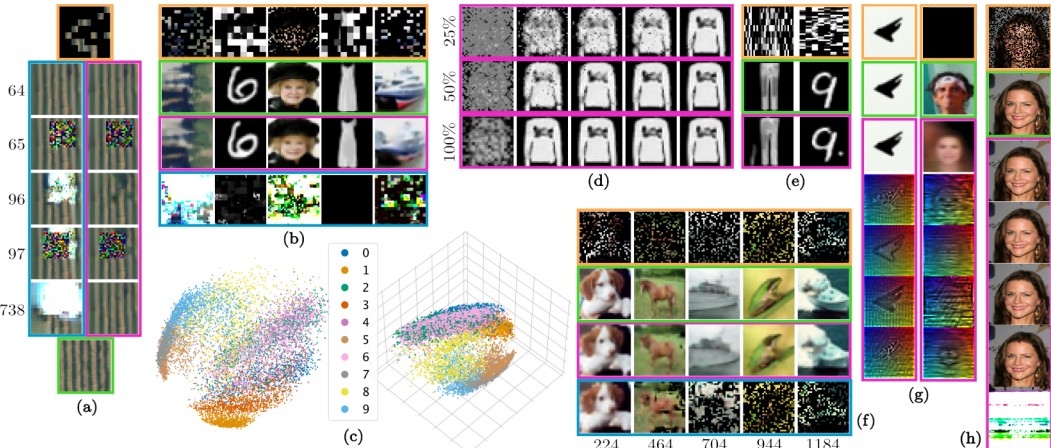

Figure 4: Qualitative results. Gold boxes are inputs, green ground truths, purple ViTCA outputs, and blue UNetCA outputs. We analyze the effects of ViTCA and UNetCA on cell states in terms of: **(a)** damage resilience; **(b)** convergence stability, and; **(c)** hidden state PCA visualizations of converged cell grids for all examples in FashionMNIST [42]. We also investigate update rule inductive biases in terms of adapting to: **(f)** varying inputs *during* cell updates; **(d)** varying cell update rates; **(e)** noise configurations unseen during training; **(g)** unmasked and completely masked inputs, and; **(h)** spatial interpolation enabled by our various methods of incorporating cell positioning.

**Convergence stability.** Fig. 4 (b) shows denoising results after 2784 cell grid updates. ViTCA is able to maintain a stable cell grid state while UNetCA causes cells to diverge.

**Hidden state visualizations.** Fig. 4 (c) shows 2D and 3D PCA dimensionality reductions on the hidden states of converged cell grids for all examples in FashionMNIST [42]. The clusters suggest some linear separability in the learned representation, motivating our probing experiments in Sec. 4.2.

### 4.1.3 Investigating update rule inductive biases

Here we investigate the inductive biases inherent in ViTCA and UNetCA by testing their adaptation to various environmental changes (Fig. 4 (d,e,f,g,h)).

**Adaptation to varying update rates.** Despite being trained with a $\sigma = 50\%$ cell update rate, ViTCA is able to adapt to varying rates (Fig. 4 (d)). Higher rates result in a proportionally faster rate of cell state convergence, and equivalently with lower rates. UNetCA exhibits a similar relationship, although is unstable at $\sigma = 100\%$ (see Appendix A.3). For details comparing training with a synchronous *vs.* asynchronous cell grid update, see Appendix A.2.

**Generalization to noise unseen during training.** ViTCA is capable of denoising configurations of noise it has not been trained on. Fig. 4 (e; *left-to-right*): $4 \times 1$ and $1 \times 4$ patches of Gaussian noise at 65% coverage. In contrast, UNetCA induces a cell state divergence (see Appendix A.3).

**Adaptation to changing inputs.** At various moments during cell updates, we re-inject cells with new masked inputs (Fig. 4 (f)). ViTCA is able to consistently adapt cells to new inputs while UNetCA experiences difficulty past a certain point (*e.g.*, at 464 iterations in the figure).

**Effects of not *vs.* completely masking input.** Fig. 4 (g; *left*): ViTCA is able to perform autoencoding despite not being trained for it. UNetCA induces a cell grid divergence (see Appendix A.3). Fig. 4 (g; *right*): Interestingly, when the input is completely masked, ViTCA outputs the median image [37]. UNetCA does not exhibit such behaviour and instead causes cells to diverge (see Appendix A.3).

**Spatial interpolation.** We use ViTCA models trained at $32 \times 32$ using various types of positioning to generate $128 \times 128$ outputs during inference, assuming an identical cell grid resolution. Fig. 4 (h; *top-to-bottom of outputs*): xy-coordinates, no positioning, Fourier features [29], Fourier features concatenated with xy-coordinates, and a Transformer-based handcrafted positional encoding (baseline) [41]. Results are ordered from best to worst. The baseline approach is not capable of spatial interpolation due to being a 1D positioning, while, as expected, the 2D encodings make it capable. Surprisingly, removing Fourier features and using only xy-coordinates results in a higher fidelity interpolation. We believe this to be caused by the distracting amount of positional information

|  |  | MNIST | | FashionMNIST | | CIFAR10 | |
| --- | --- | --- | --- | --- | --- | --- | --- |
|  |  | Acc. ↑ | # Params. | Acc. ↑ | # Params. | Acc. ↑ | # Params. |
| **Baselines** | U-Net | 96.3 | 15.4K | 86.2 | 15.4K | 52.3 | 15.4K |
|  | ViT | 92.1 | 1.3M | 83.4 | 1.3M | 34.5 | 1.3M |
|  | UNetCA | 96.3 | 327.7K | 89.5 | 327.7K | **55.1** | 327.7K |
|  | ViTCA | 96.7 | 327.7K | 89.7 | 327.7K | 50.2 | 327.7K |
| **Variants** | ViTCA-32 | 96.3 | 327.7K | 89.8 | 327.7K | **55.1** | 327.7K |
|  | ViTCA-32xy | 96.3 | 327.7K | 89.5 | 327.7K | 53.6 | 327.7K |
|  | ViTCA-i | 95.8 | 327.7K | 89.6 | 327.7K | 49.4 | 327.7K |
|  | ViTCA-i16 | 95.7 | 327.7K | **90.1** | 327.7K | 50.7 | 327.7K |
|  | ViTCA-ixy | 96.2 | 327.7K | 89.6 | 327.7K | 50.2 | 327.7K |
|  | ViTCA-i16xy | 96.5 | 327.7K | 89.6 | 327.7K | 52.7 | 327.7K |
|  | Linear classifier | 93.0 | 10.3K | 84.7 | 10.3K | 39.0 | 30.7K |
|  | 2-layer MLP, 100 hidden units | 98.2 | 103.5K | 89.4 | 103.5K | 46.0 | 308.3K |
|  | 2-layer MLP, 1000 hidden units | **98.5** | 1.0M | 89.6 | 1.0M | 49.7 | 3.1M |

Table 3: Linear probe [28] test accuracies (%) of baseline and variant models. Model variants are labelled as in Tab. 1. All baselines and variants were pre-trained for denoising autoencoding and kept fixed during probing. A linear classifier and 2-layer Multilayer Perceptrons (MLP) were trained on raw image inputs. Parameter counts exclude fixed parameters. Boldface and underlined values denote the best and second best results, respectively. Interestingly, CA-based models trained for denoising autoencoding on increasingly challenging datasets produce an increasingly more useful self-supervised representation for image classification compared to non-CA-based models.

Fourier features provide to cells, as cells can instead rely on their hidden states to store higher frequency positional information. Finally, with no explicit positioning, ViTCA is still able to perform high-quality interpolation—even exceeding using Fourier features—by taking advantage of its self-organizing nature. As a side note, we point attention to the fact that ViTCA is simultaneously denoising at a scale space it has not been trained on, exemplifying its generalization capabilities.

## 4.2 Investigating hidden representations via linear probes

Here we examine the learned representations of our models pre-trained for denoising. We freeze model parameters and learn linear classifiers on each of their learned representations: converged cell hidden states for CA-based models, bottleneck features for U-Net, and LN'd tokens for ViT. This is a common approach used to probe learned representations [28]. Classification results on MNIST, FashionMNIST, and CIFAR10 are shown in Tab. 3 and we use the same training setup as for denoising, but without any noise. For comparison, we also provide results using a linear classifier and two 2-layer MLPs of varying complexity, all trained directly on raw pixel values. Correlations between denoising performance in Tab. 1 and classification performance in Tab. 3 can be observed. Linear classification accuracy on ViTCA-based features typically exceeds classification accuracy using other model-based features or raw pixel values, even outperforming the MLPs in most cases.

## 5 Discussion

We have performed extensive quantitative and qualitative evaluations of our newly proposed ViTCA on a variety of datasets under a denoising autoencoding framework. We have demonstrated the superior denoising performance and robustness of our model when compared to a U-Net-based CA baseline (UNetCA) and ViT, as well as its generalization capabilities under a variety of environmental changes such as larger inputs (*i.e.*, spatial interpolation) and changing inputs *during* cell updates.

Despite the computation savings—owed to our circumvention of self-attention's quadratic complexity by spatially localizing it within ViTCA—there remains the same memory limitations inherent to all recurrent models: multiple recurrent iterations are required for each training iteration, resulting in larger memory usage than a feedforward approach. This limits single-GPU training accessibility. We have experimented with gradient checkpointing [44] but found its trade-off for increased back-propagation duration (and slightly different gradients) less than ideal. To fully realize the potential of NCAs (self-organization, inherent distributivity, *etc*.), we encourage follow-up work to address this limitation. Adapting recent techniques using implicit differentiation is one avenue to circumvent these issues [2, 32]. Also, as mentioned in our ablation (Sec. 4.1.1), we hope to further investigate the instabilities caused by increasing the depth of ViTCA.

## Acknowledgments and disclosure of funding

First and foremost, M.T. thanks their former supervisor and mentor, Konstantinos (Kosta) G. Derpanis, for his invaluable support throughout the project. M.T. also thanks Martin Weiss for his helpful feedback on implementing the linear probe experiments (Sec. 4.2); Olexa Bilaniuk for his assistance in investigating the gradient differences caused by PyTorch's gradient checkpointing implementation (see Appendix A.2), and; the Mila Innovation, Development, and Technology (IDT) team for their overall technical support, particularly, their tireless efforts maintaining cluster reliability during the crucial moments preceding the submission deadline. The authors acknowledge the material support of NVIDIA in the form of computational resources.

M.T. is partially supported by the Natural Sciences and Engineering Research Council of Canada (NSERC) Canada Graduate Scholarship – Doctoral [application number CGSD3-519428-2018]. D.N. and C.P. are each partially supported by a NSERC Discovery Grant [application IDs 5011360 and 5018358, respectively]. D.N. thanks Samsung Electronics Co. Ltd. for their support. C.P. thanks CIFAR for their support under the AI Chairs Program.

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
