# OpenReview forum: "Attention-based Neural Cellular Automata"
_NeurIPS.cc/2022/Conference — NeurIPS 2022 Accept_

### Official Review · Reviewer_mu6v · 2022-07-08

**Rating:** 7
**Confidence:** 4
**Soundness:** 2 fair
**Presentation:** 3 good
**Contribution:** 2 fair

**Summary:**

The paper presents a neural cellular automata (NCA) architecture for image denoising that replaces the typical convolutional layers with a vision transformer.

The experiments show that the proposed architecture achieves marginally better results compared to two baselines on three simple datasets, and the authors show that their model appears to be stable after converging to a target image (NCAs are known to be unstable when evaluated on longer horizons than the one seen at training time).

**Questions:**

Overall, I lean towards a borderline reject which can turn into a borderline accept if the following questions are addressed.

1. What is the meaning of "it is a fairly contractive model capable of fixed-point convergence guarantees"?

2. The attention operation in Equations (1) and (3) is at the patch level, not at the cell level. This means that ViTCA is not comparable 1-to-1 with pixel-level NCAs unless the patch size is set to 1x1.
Also, it might confuse the reader to refer to the "Moore neighborhood" in line 177 when talking about the patch size, because this term is typically used in CA literature to indicate the neighborhood of a cell (as it is used to compute the update). Here, the effective neighborhood of the cell is 9x9.
Did I misunderstand something?

3. I couldn't find how the positional encodings are computed. IF PEs provide unique coordinates for each pixel, doesn't that defeat the principle of locality of CAs?

4. Why is the ViTCA baseline configured to have a patch size of 1x1, while all the other ViTCA models have 3x3?

5. The number of parameters in Tables 1, 2, and 3 appears to be misreported.
A few examples:
    - Table 1, no difference between ViTCA (4 heads) and ViTCA-32 (32 heads)
    - Table 2, no difference between ViTCA with 4-64 heads
    - Table 3, no difference between the variants
    - Table 3, no difference between MLPs with 100 or 1000 hidden units on CIFAR10

    The absence of differences in Tables 1 and 2 is especially puzzling and important to clarify, because ViTCA-32 is the only model that can consistently surpass the baselines (while, e.g., ViTCA-4 cannot).

6. Why use an overflow loss instead of an activation function (sigmoid, tanh) that forces the updates to be in the required ranges?

7. Figure 3 shows some noise sampled from a uniform distribution being injected in the cell update. Where is this described in the text?

8. Since all NCAs are image-to-image maps, why didn't the authors consider other NCA baselines like the original NCA [30] or the Variational NCA [7]? Why not consider other recurrent baselines too? As it is (and considering point 5 above), the comparisons are not very informative.

I'm looking forward to the discussion with the authors.

**Limitations:**

The limitations are not discussed extensively besides what concerns memory costs.
I have pointed out a few issues in my questions that could be discussed better (e.g., point 3).

There are no societal concerns with the work.

**Strengths And Weaknesses:**

The idea of the paper is novel, although it doesn't provide substantial contributions besides replacing convolutions with transformers.
The quality of the paper is good, the experiments go fairly in-depth although on just one task of image denoising (which is less interesting than texture generation or other forms of morphogenesis, in the reviewer's personal opinion).

The clarity of the paper is good, almost all details are reported clearly and there are only a few details that require clarification (see below).
The significance of the paper is modest, since it is not even clear that the proposed method provides real advantages compared to the two simple baselines tested by the authors. The paper is unlikely to have an impact on the NCA community, and surely not on the much larger computer vision community.

---

> ### Author Response · Authors · 2022-08-02
> **Response to Reviewer mu6v (pt. 1)**
>
> We thank you for your extensive feedback and for recognizing the novelty of our idea. We are particularly grateful for your keen eye in pointing out the mistake in Table 3 on the MLP parameter counts. We have made a correction and used the opportunity to clarify the meaning of the parameter counts in the caption. Below we have provided our responses to your questions and to some of your comments. We hope we left "no stone unturned" with our responses, but if we have, we wholly welcome and look forward to more feedback. Finally, we have also added a top-level comment to summarize the changes made in response to all four reviewers' comments and questions.
>
> **Statement**: _The experiments show that the proposed architecture achieves marginally better results compared to two baselines on three simple datasets..._
>
> **Author response**: We are confused with this statement because our denoising autoencoding experiments use six (not three) datasets, including CelebA and Tiny ImageNet which are not generally considered simple datasets. We want to clarify that Vision Transformers have yet to achieve competitive performance against U-Nets (and convolutional networks in general) on image synthesis-based tasks, which require efficient pixel-level processing. Our work presents an architecture which simply envelops ViT within an NCA paradigm, with little change to the underlying ViT (the only notable change being localized attention) architecture, resulting in a complete transformation in ViT's performance on image synthesis-based tasks. To the point where it becomes competitive against a U-Net having more parameters. Although the number of parameters in ViTCA is around 10% more than in ViT, this is only due to the dimensionality difference in the inputs and outputs, since ViTCA's inputs and outputs consist of high dimensional cells instead of _just_ image patches. However, the performance gap between ViTCA and ViT averages in the double digits. For the sake of completion, below we provide results comparing an inverted bottleneck ViT (which we did not provide in the paper but did have some results on) with ViTCA-i in case the argument is made that the comparison between ViT and ViTCA is unfair due to parameter count differences (the parameter count differences between ViT-i and ViTCA-i is smaller than between ViT and ViTCA). Specifically, we provide four tables of denoising autoencoding results on the test sets for LandCoverRep, CelebA, MNIST, and FashionMNIST. We did not have results on CIFAR-10 or TinyImageNet due to time constraints.
>
> |         |  PSNR |  SSIM | LPIPS | # Params. |
> |:-------:|:-----:|:-----:|:-----:|:---------:|
> | VITCA-i | **33.49** | **0.929** | **0.108** |   54.7K   |
> | ViT-i   | 29.99 | 0.878 | 0.154 |   50.4K   |
>
> **Table 1**: Comparing denoising autoencoding test results on LandCoverRep between inverted bottleneck variants of ViT and ViTCA.
>
>
> |         |  PSNR |  SSIM | LPIPS | # Params. |
> |:-------:|:-----:|:-----:|:-----:|:---------:|
> | VITCA-i | **26.10** | **0.904** | **0.074** |   54.7K   |
> | ViT-i   | 18.87 | 0.737 | 0.310 |   50.4K   |
>
> **Table 2**: Comparing denoising autoencoding test results on CelebA between inverted bottleneck variants of ViT and ViTCA.
>
>
> |         |    PSNR   |    SSIM   |   LPIPS   | # Params. |
> |:-------:|:---------:|:---------:|:---------:|:---------:|
> | VITCA-i | **26.03** | **0.930** | **0.033** |   54.3K   |
> | ViT-i   | 14.76     | 0.537     | 0.313     |   50.1K   |
>
> **Table 3**: Comparing denoising autoencoding test results on MNIST between inverted bottleneck variants of ViT and ViTCA.
>
>
> |         |    PSNR   |    SSIM   |   LPIPS   | # Params. |
> |:-------:|:---------:|:---------:|:---------:|:---------:|
> | VITCA-i | **22.84** | **0.827** | **0.139** |   54.3K   |
> | ViT-i   | 15.64     | 0.502     | 0.405     |   50.1K   |
>
> **Table 4**: Comparing denoising autoencoding test results on FashionMNIST between inverted bottleneck variants of ViT and ViTCA.
>
> Although ViTCA-i has **~8%** more parameters than ViT-i (due to differences in input/output dimensionality), when considering that the scale of PSNR is logarithmic, that the range of SSIM is [0, 1], and that the range of LPIPS is [0, 1], there is a substantial jump in performance in ViTCA-i compared to ViT-i. On average, there is a **43.1%** relative improvement in PSNR, a **23.4%** absolute improvement in SSIM, and a **20.7%** absolute improvement in LPIPS.

---

> > ### Author Response · Authors · 2022-08-02
> > **Response to Reviewer mu6v (pt. 2)**
> >
> > **Statement**: _The idea of the paper is novel, although it doesn't provide substantial contributions besides replacing convolutions with transformers._
> >
> > **Author response**: We are happy to see that you recognize the novelty of the idea. The integration of the spatially-localized attention mechanism within the NCA framework is not straightforward, and there are many pivotal points when designing such a mechanism where certain design choices can lead to pitfalls. When introducing an architecture that marries two such seemingly incompatible frameworks, we feel it is critical to explore architectural variants and other design choices in-depth to verify that the results from the baseline implementation of the core idea were not spurious. Thus, we decided to choose a well-generalizeable image-based task as our vehicle for exploring our core idea as opposed to exploring multiple tasks, which can be left for future work. In doing so, we have provided extensive ablative results exploring modifications such as:
> >
> > - Transformer embedding dimensionality
> > - The number of attention heads
> > - Transformer depth
> > - Cell hidden dimensionality size
> > - Cell initialization methods
> > - Inverted bottlenecking
> > - The effects of posititional encoding
> > - and more...
> >
> > Furthermore, in combining ViTs with NCAs, we were confronted with exploring the effects of positional encodings in a NCA framework, which has not been explored previously. In relation to your third question, we had an initial hesitancy in incorporating positional information in a NCA framework but realized that the restriction pertaining to locality in CAs is just a restriction on the locality of computation and not that the cells can not contain their own explicit positioning. Please refer to our answer to your third question for more.

---

> > > ### Author Response · Authors · 2022-08-02
> > > **Response to Reviewer mu6v (pt. 3)**
> > >
> > >
> > > **Q1**: _What is the meaning of "it is a fairly contractive model capable of fixed-point convergence guarantees"?_
> > >
> > > **Author response**: The full sentence in question (L58-60):
> > >
> > > "Since ViT (and by extension, ViTCA) employs Layer Normalization (LN) [43] at each stage of its processing, it is a fairly contractive model capable of fixed-point convergence guarantees [32]."
> > >
> > > This sentence cites the Deep Equilibrium Models (DEQ) paper [32] which is about analyzing recurrent models (RNNs), observing how their hidden states transform over recurrent updates, and analytically describing how at the limit of time the hidden states converge to an "equilibrium point," or synomously, a "stable" or "fixed" point. Inspired by the Neural ODE paper [1] (reference provided below, not to be mixed up with the paper's references), they decide that instead of recurrently updating until the fixed point is reached, they can use out-of-the-box fixed-point solvers which treat the problem of finding a fixed-point as a root finding problem (_i.e.,_ when the change in state is 0). They highlight some recurrent models to test their theory, one of them being a recurrent autoregressive Transformer for next-symbol prediction in natural language processing. Importantly, near the bottom of page 8 in their paper, they highlight how the Layer Norms in the Transformer they use help make it "contractive (and stable)" which is important for it to be able to transform hidden states to a stable equilibrium point.
> > >
> > > Vision Transformers (and Transformers) use Layer Norm to transform activation statistics for each token to be standardized (zero mean, unit standard dev). In ViT, it's applied before each attention block, before each feedforward MLP block, and before the final MLP head. Furthermore, there are learned affine parameters that can scale and translate the normalization as the network sees fit. The result is that the transformation of each token (which are cells in ViTCA's case) is more stable (fewer erratic and drastic changes to tokens as you process a sequence of tokens), predictable, and thus the model is easier to train. Furthermore, in a recurrent setting, when recurrently transforming the same tokens over time, it combats against explosive growth in token values that can occur due to any unchecked activations or operators that, without the normalization, would result in an expansive mapping (i.e., a Transformer that is not Lipschitz continuous).
> > >
> > > For better understanding the "contractive" part of our sentence, we point to the Banach fixed-point theorem which states that if a mapping $F: \mathbf{Z} \mapsto \mathbf{Z}$ is a contraction (a special kind of Lipschitz continuity where the Lipschitz constant $K$ is between $0 \leq K \lt 1$ and $F$ maps a metric space to itself) then repeated applications of $F$ on $\mathbf{Z}$, like so: $\mathbf{Z}, F(\mathbf{Z}), F(F(\mathbf{Z})), ...$ will result in a convergence to a unique fixed point at the limit. In other words, it will result in a stable result that does not change after more repeated applications of $F$. This is perhaps one of the reasons why ViTCA is much more capable of keeping cells at their converged state compared to UNetCA, despite the two of them being trained with an overflow loss. We also experimented with training ViTCA without Layer Normalization and although it was capable of stabilizing after convergence during test time on an example, the quality of the results were not as good. We are excited to explore the application of DEQs within ViTCA and NCAs in general.

---

> > > > ### Author Response · Authors · 2022-08-02
> > > > **Response to Reviewer mu6v (pt. 4)**
> > > >
> > > >
> > > >
> > > > **Q2**: _The attention operation in Equations (1) and (3) is at the patch level, not at the cell level. This means that ViTCA is not comparable 1-to-1 with pixel-level NCAs unless the patch size is set to 1x1. Also, it might confuse the reader to refer to the "Moore neighborhood" in line 177 when talking about the patch size, because this term is typically used in CA literature to indicate the neighborhood of a cell (as it is used to compute the update). Here, the effective neighborhood of the cell is 9x9. Did I misunderstand something?_
> > > >
> > > > **Author response**: Yes, there is a misunderstanding here. We believe your are confusing patch size with cell neighbourhood size. You are correct in stating that Eq. 1 and Eq. 3 operate at the patch level. However, this is _also_ the cell level. As shown in L119, the "flattened cell grid" $\mathbf{Z}$ has dimensions $N \times L$ where $N$ is the number of cells and $L = C_PP_HP_W + C_h$ is the dimensionality of each cell. Note the $P_HP_W$ which imply that each cell contains an input image patch. Figure 3 and its caption explicitly describe this.
> > > >
> > > > Unless otherwise stated, all models which perform input image patchification (_i.e.,_ ViT baseline and variants and ViTCA baseline and variants) do so with a patch size of $P_H \times P_W = 1 \times 1$, which is equivalent to not doing any patchification at all since $1\times1$ is pixel-sized. We have modified L175 to be more clear in explaining this. The only experiment where patch size differs is in the patch size ablation in Appendix A.2, where ViTCA's patch size is modified.
> > > >
> > > > The Moore neighbourhood described in L177 is used for describing the $3\times3$ cell neighbourhood size, which is synonomous with the update rule's attention neighbourhood size. L177 describes the Moore neighbourhood size as being $N_H \times N_W = 3\times3$, with $N_H$ and $N_W$ being referenced in L143-147 to describe the spatially localized attention window size. Furthermore, the attention neighbourhood size ablation in L580-588 in Appendix A.2 gets into more detail explaining the relationship between performance and local attention size while explicitly relating the Moore neighbourhood to the $3\times3$ attention size used by the baseline ViTCA and other variants (L586-588).

---

> > > > > ### Author Response · Authors · 2022-08-02
> > > > > **Response to Reviewer mu6v (pt. 5)**
> > > > >
> > > > >
> > > > > **Q3**: _I couldn't find how the positional encodings are computed. IF PEs provide unique coordinates for each pixel, doesn't that defeat the principle of locality of CAs?_
> > > > >
> > > > > **Author response**: Descriptions for how the positional encodings are computed are in L229-253. We have corrected a mistake in L244 where "(1b) xy..." should be "(1c) xy..."
> > > > >
> > > > > To clarify, there are two groups of methods of incorporating positional information that we experiment with:
> > > > >
> > > > > 1. Injecting/concatenating absolute (xy) or relative (Fourier-based) coordinates into each cell ($\gamma$ in Figure 3) which the update rule can not modify. You can call this the NeRF style of coordinate-based positioning:
> > > > >
> > > > >     (a) _sincos5_: Handcrafted Fourier features computed as described in L237-238.
> > > > >
> > > > >     (b) _sincos5xy_: Handcrafted Fourier features computed as described in L237-238 but with xy-coordinates appended as well.
> > > > >
> > > > >     (c\) _xy_: Just xy-coordinates where each cell's coordinate in the grid is appended to it.
> > > > > 2. Using the Transformer-style of positional encoding/embedding where it's added into the embedding:
> > > > >
> > > > >     (a) _handcrafted_: Handcrafted Fourier features which are not computed the same way as with (1a) although the two share similarities. The way this handcrafted Transformer-style of positional encoding is computed is described in [41] and is also how Vision Transformers compute positional encodings.
> > > > >
> > > > >     (b) _learned_: Instead of using a handcrafted positional encoding, a learned positional _embedding_ is added into the cell embeddings. This was experimented in the original Vision Transformer paper [13] and they found it too limiting since it restricts the input/output size and so they stuck with the above handcrafted approach.
> > > > >
> > > > > Injecting positioning has an effect on cell dimensionality since it is a concatenation, which in turn has an effect on the total number of trainable parameters.
> > > > >
> > > > > "_IF PEs provide unique coordinates for each pixel, doesn't that defeat the principle of locality of CAs?_"
> > > > >
> > > > > You raise a valid concern. However, we believe you are conflating self-organization with self-localization. The principle of locality of CAs is based on the local computational aspect of the update rule (L61-62), not on whether the cells it's operating on contain positional information. We are not aware of any CA literature which explicitly states a limitation on providing positional information or which require cells to even be able to localize themselves. As long as the update rule is operating on a neighbourhood of cells, it is valid. We would like to point out that just because a cell knows its relative or absolute position, it does not mean that it can use that information to immediately know the state of cells past its neighbours. That information must still be propagated over time, _i.e.,_ the update rule is still operating on local information which is spatially propagated over time so that future cell updates will implicitly be a function of not only the positions of cells within a neighbourhood but also the positions of cells past its neighbourhood. This is no different than the situation where no explicit positioning is provided and when it's hidden instead in the cell hidden channels. We also show that we don't need explicit positioning for the model to work and that although it doesn't perform as well compared to the baseline ViTCA, it's still quite close. This is in contrast to ViT without positioning, where our early experiments showed it fails miserably with no coherent output at all.
> > > > >
> > > > > As a closing note, we feel that this idea of providing positional information to cells can be an exciting opportunity to investigate a case where cells are provided with initial positional information and are tasked to refine this information to better suit their needs depending on the task and dataset being trained on, inspired by this paper [2] (reference at the bottom). At that point one must wonder whether cells are moving themselves around in the grid or if they are remaining fixed but moving themselves in their "imagined" representation of their spatial surroundings. It's an interesting thought experiment.

---

> > > > > > ### Author Response · Authors · 2022-08-02
> > > > > > **Response to Reviewer mu6v (pt. 6)**
> > > > > >
> > > > > >
> > > > > >
> > > > > > **Q4**: _Why is the ViTCA baseline configured to have a patch size of 1x1, while all the other ViTCA models have 3x3?_
> > > > > >
> > > > > > **Author response**: We are wondering if this is a misunderstanding stemming from L177 where the Moore neighbourhood is brought up. Unless otherwise stated, all models which perform input image patchification (_i.e.,_ ViT baseline and variants and ViTCA baseline and variants) do so with a patch size of $P_H \times P_W = 1 \times 1$, which is equivalent to not doing any patchification at all since $1\times1$ is pixel-sized. We have modified L175 to be more clear in explaining this.
> > > > > >
> > > > > >
> > > > > > **Q5**: _The number of parameters in Tables 1, 2, and 3 appears to be misreported. A few examples:_
> > > > > > * _Table 1, no difference between ViTCA (4 heads) and ViTCA-32 (32 heads)_
> > > > > > * _Table 2, no difference between ViTCA with 4-64 heads_
> > > > > > * _Table 3, no difference between the variants_
> > > > > > * _Table 3, no difference between MLPs with 100 or 1000 hidden units on CIFAR10_
> > > > > >
> > > > > > _The absence of differences in Tables 1 and 2 is especially puzzling and important to clarify, because ViTCA-32 is the only model that can consistently surpass the baselines (while, e.g., ViTCA-4 cannot)._
> > > > > >
> > > > > > **Author response**: Below we provide our responses to each of the bullet points mentioned in your question.
> > > > > > * _Table 1, no difference between ViTCA (4 heads) and ViTCA-32 (32 heads)_
> > > > > > * _Table 2, no difference between ViTCA with 4-64 heads_
> > > > > >
> > > > > > Since we follow ViT's formulation of multi-head self-attention (MHSA), the parameter complexity of the model is kept constant as more heads are added. This is evidenced on L130 where $d$ (embed size) is divided by $h$ (number of heads). However, when $h = 1$ (single-head self-attention), there are fewer parameters than when $h = 2$ because single-head self-attention lacks the final weight matrix that's used to combine multiple heads before adding the result into the embedding. Although this explains why ViTCA and ViTCA-32 have the same number of parameters, it does not mean that adding more heads in ViTCA results in free performance gains. Due to the lack of low-level CUDA optimizations for attention-based operations (unlike with convolutions), more heads results in more computation time and more memory usage. This is made even more complicated when spatial localization is involved. We have managed to optimize our spatially-localized MHSA implementation when only using PyTorch's standard library (even investigating JIT compiling with TorchScript), but we are not completely satisfied yet since we know it can still be improved. We hope to implement our own CUDA kernels for our spatially-localized MHSA in the future.
> > > > > >
> > > > > > * _Table 3, no difference between the variants_
> > > > > >
> > > > > > This is a valid misunderstanding. The parameter counts listed in Table 3 are for the number of trainable parameters. Since linear probing is done with frozen/fixed pre-trained models, this means the parameters listed are the number of parameters for the linear classifier. This was an oversight on our part and has been clarified in the caption of Table 3 in the revised manuscript.
> > > > > >
> > > > > > * _Table 3, no difference between MLPs with 100 or 1000 hidden units on CIFAR10_
> > > > > >
> > > > > > Thank you for pointing this out. To succinctly answer, the 1000 hidden units linear classifier should have 3.1M parameters. This does not change the overall narrative though and in fact puts it more in favour of our model.

---

> > > > > > > ### Author Response · Authors · 2022-08-02
> > > > > > > **Response to Reviewer mu6v (pt. 7)**
> > > > > > >
> > > > > > >
> > > > > > > **Q6**: _Why use an overflow loss instead of an activation function (sigmoid, tanh) that forces the updates to be in the required ranges?_
> > > > > > >
> > > > > > > **Author response**: We believe there is a misunderstanding of the purpose of the overflow loss. The overflow loss is not meant for restricting the range of cell update values, but rather for encouraging the range of cell _output_ values to be within [0, 1] and cell _hidden_ values to be within [-1, 1]. It is intended to encourage long term cell state stability, inspired by [20] and [22] (in the paper's references). On the other hand, the cell _updates_ produced by the update rule, which _update cell values_, are unbounded. This allows for additive and subtractive cell updates, as done by [20], [21], and [30]. Despite this, we _did_ experiment with restricting the bounds of cell updates by using a sigmoid, tanh, or ReLU activation after the MLP head's linear layer (which produces the update each iteration).
> > > > > > >
> > > > > > > Before explaining our results for each, it's important to highlight the tricks we implemented to keep training stable and manageable:
> > > > > > >
> > > > > > > First, during training we zero initialize the weights and biases of the final layer of our CA-based models (mentioned in L199) which allows for the update rule to exhibit a "do nothing" initial behaviour (this idea was taken from [30]). Since cell output channels are initialized to 0.5, cell hidden channels are initialized to 0, target image RGB values are in the range [0, 1], and overflow loss occurs when the output RGB values are past the range [0, 1] and when the hidden values are past the range [-1, 1], this results in the first CA training iteration having a loss with low magnitude. This means gradients are also of low magnitude and the second training iteration will not result in a dramatic change in the loss. This makes the training process more stable. We experimented with initializing the weights and biases of the final layer the same as with other layers (He initialization) and noticed the training process quickly grow unstable, which was most unstable with UNetCA since it lacked any normalization layers like ViTCA does.
> > > > > > >
> > > > > > > Second, ViTCA's layer normalizations, which helps keep internal activations within a manageable and stable range, which in turn improves training stability.
> > > > > > >
> > > > > > > Third, before applying gradients accumulated after the $T$ recurrent cell updates, we normalize them. Line 19 in Algorithm 1 (in Appendix A) shows this.
> > > > > > >
> > > > > > > Now we can get into the results of our activation experiments:
> > > > > > >
> > > > > > > _Sigmoid:_
> > > > > > >
> > > > > > > This eliminated the "do nothing" initial behaviour since it transformed the zero output of the final layer (since its weights and biases were initialized to 0) to a 0.5 output, which resulted in unstable training by immediately increasing cell values. Furthermore, since a sigmoid outputs values between (0, 1), this meant cell updates were always additive and were never subtractive, which caused explosive cell state growth and unstable training. On a related note, recurrent models are much more susceptible to the exploding gradient problem when using sigmoid activations. This typically leads to using ReLUs instead.
> > > > > > >
> > > > > > > _ReLU:_
> > > > > > >
> > > > > > > As experienced by [30], using a ReLU resulted in only positive cell updates and no subtractive cell updates. Also, since it's unbounded, it quickly resulted in explosive cell state growth, which destabilized training.
> > > > > > >
> > > > > > > _Tanh:_
> > > > > > >
> > > > > > > Tanh allowed for both additive and subtractive cell state updates. Although its derivative quickly shrinks with non-zero inputs (even with input values with an absolute value < 1)---which can cause vanishing gradients---our use of gradient normalization, our weight/bias initialization (He init except for final layer that uses zero init), the fact that target RGB values are in the range [0, 1], and initial cell output values being 0.5 meant that vanishing gradients wouldn't be much of a problem, at least during the start of training when inputs to tanh were small. The problem with using tanh was that it was still susceptible to vanishing gradients _over time_, despite our mitigations, and it slowed down training progress since it would prevent being able to quickly update cells to the desired ranges (which would also result in worse training losses). We found it preferable to just not use an activation function since the training process was already quite stable.
> > > > > > >
> > > > > > >
> > > > > > > **Q7**: _Figure 3 shows some noise sampled from a uniform distribution being injected in the cell update. Where is this described in the text?_
> > > > > > >
> > > > > > > **Author response**: The $\sigma$ shown in Figure 3 is used for asynchronous updating. It is not being injected in the cell update, it is being multiplied with the cell update, hence the Hadamard product symbol $\odot$ that is placed between $\sigma$ and the cell update.

---

> > > > > > > > ### Author Response · Authors · 2022-08-02
> > > > > > > > **Response to Reviewer mu6v (pt. 8)**
> > > > > > > >
> > > > > > > >
> > > > > > > > **Q8**: _Since all NCAs are image-to-image maps, why didn't the authors consider other NCA baselines like the original NCA [30] or the Variational NCA [7]? Why not consider other recurrent baselines too? As it is (and considering point 5 above), the comparisons are not very informative_
> > > > > > > >
> > > > > > > > **Author response**: Although all NCAs presented so far have been image-to-image maps, it is not a necessity. One can imagine an NCA operating on audio waveforms where cells are connected along a one dimensional line, or an NCA operating on sentences for next word prediction. Despite this, we acknowledge the validity of your question and provide our answer.
> > > > > > > >
> > > > > > > > _Why not the original NCA?_
> > > > > > > >
> > > > > > > > The original NCA from the Growing NCA distill publication [30] has around 8K parameters and was designed for "growing" an image from a single seed cell. It consists of an initial perception stage using three fixed filters (sobel_x, sobel_y, identity) followed by two fully connected layers separated by a ReLU. We felt it would not make sense to compare such a small NCA designed with such a different task in mind. Particularly, it used "living cell masking" and most importantly did not employ any input injection, _i.e.,_ it was not a conditional model trained on a dataset. One can artificially bump up its parameter count or lower ViTCA's parameter count but there are still fundamental differences in the architecture that led us to believe it would not perform well for denoising autoencoding. Conversely, we felt that including another task, specifically the task of growing an image from a single seed with living cell masking, would not have been as informative as denoising autoencoding when inferring the generalizability of ViTCA on other image synthesis-based tasks, such as texture synthesis. On that note, the follow up to [30] was on texture synthesis ([20] and [21]) and it was in fact the first task we started with when testing ViTCA. We did not include our results on texture synthesis since it would have taken up space in the main manuscript that we wanted to occupy with our extensive ablations. Texture synthesis, at least how it was implemented in [20] and [21], is a much more specialized task than denoising autoencoding in the sense that there is no "training" on a dataset, instead it is an optimization on a single examplar texture. Also, it lacks the higher level semantics present in images like faces, vehicles, and animals, meaning it is focused on a more specific kind of image representation unlike with denoising autoencoding. Despite all this, we are delighted to share some results we had from these early experiments, linked in this [imgur link](https://imgur.com/a/5JwVHzW). To sum it up, compared to Texture ViTCA, Texture NCA takes a much longer time to output a texture which looks qualitatively worse, all while using 44% more parameters than Texture ViTCA. We hope to investigate this direction in the future, especially considering the exciting advancements made in [20] with its Optimal Transport Texture loss.
> > > > > > > >
> > > > > > > > _Why not Variational NCA?_
> > > > > > > >
> > > > > > > > Variational NCA is a generative model while our architecture is a discriminative model. Furthermore, the contributions vastly differ between our paper and VNCA. Our contributions are centred on making global self-attention work in a strictly local computational model. Variational NCA's contributions are centred on introducing probabilistic generative modeling within an NCA framework, specifically using variational inference. Our approach is not meant to compete with Variational NCA and can actually be combined with Variational NCA to produce a probabilistic generative ViTCA, which would be an exciting direction for future work.

---

> > > > > > > > > ### Author Response · Authors · 2022-08-02
> > > > > > > > > **Response to Reviewer mu6v (pt. 9)**
> > > > > > > > >
> > > > > > > > > _Why not other recurrent models?_
> > > > > > > > >
> > > > > > > > > This is a reasonable question. We wanted to remain focused on exploring the NCA paradigm and to not distract ourselves on chasing the state-of-the-art. We set out a goal to explore the benefits (and short-comings) of implementing localized attention within an NCA paradigm via a Vision Transformer. Initially, we felt it was perplexing to incorporate a global computation mechanism (global self-attention) within a local computation framework (NCAs) without breaking important rules that pertained to both ViTs and NCAs. However, we managed to find a way and realized, ironically, how well suited ViTs are within an NCA paradigm because global self-attention information can be made implicit through the hidden information stored in cells and slowly aggregated over time through repeated local self-attention operations. By designing our research goals this way, it meant that we had to include ViT as a baseline model and a non-ViT-based NCA that was powerful enough to serve as a fair comparison to ViTCA. Since we wanted to focus on analyzing ViTCA's inductive biases rather than its capabilities on a variety of tasks, we chose denoising autoencoding as our relatively generic image synthesis-based task. With all of this considered, we included U-Net since it is a reliable and widely-used competitive baseline for all sorts of image-based tasks, including image synthesis and image classification. From here, naturally, we had to include a UNetCA baseline. In the interest of time and keeping our narrative focused, we decided against including any other types of baselines.
> > > > > > > > >
> > > > > > > > > **References**
> > > > > > > > >
> > > > > > > > > [1] R. T. Q. Chen, Y. Rubanova, J. Bettencourt, and D. K. Duvenaud, "Neural ordinary differential equations," in _Neural Information Processing Systems (NeurIPS)_, 2018.
> > > > > > > > >
> > > > > > > > > [2] X. Liu, H.-F. Yu, I. Dhillon, and C.-J. Hsieh, "Learning to encode position for transformer with continuous dynamical model," in _International Conference on Machine Learning (ICML)_, 2020.

---

> > > > > > > > > > ### Comment · Reviewer_mu6v · 2022-08-03
> > > > > > > > > > **Reply to authors**
> > > > > > > > > >
> > > > > > > > > > I thank the authors for their very thorough reply which has indeed helped me understand the paper better and change my mind about it.
> > > > > > > > > >
> > > > > > > > > > A few comments about the reply:
> > > > > > > > > >
> > > > > > > > > > Parts 1 and 2: Please disregard my comment on the three datasets, it was an oversight on my part and I apologize for it.
> > > > > > > > > > Indeed this paper shows that NCA-like architectures can provide significant benefits to ViTs, which are still somewhat underwhelming compared to U-nets, and this is an interesting result.
> > > > > > > > > >
> > > > > > > > > > Part 3: Thanks for clarifying. I agree with the statement, I was just confused by the wording (still, I don't think that a model can be "capable of [...] guarantees" -- maybe it can be said better).
> > > > > > > > > >
> > > > > > > > > > Part 4: Thanks for clarifying.
> > > > > > > > > >
> > > > > > > > > > Part 5: While I agree in principle, especially if the goal of the NCA is "just" denoising and not modelling biological systems, I am not aware of CA models that explicitly take into account positional encodings (I do believe that self-localization, particularly in NCA, is what makes them interesting to some degree -- otherwise a morphogenesis NCA could simply overfit the map from PE to pixel output).
> > > > > > > > > > However, I see the authors' point and I agree that it might be an interesting direction of research for the future (I especially enjoyed the closing remark). Plus, here PEs have a clear function and provide a non-trivial advantage, so they're well justified. Thanks for the discussion.
> > > > > > > > > >
> > > > > > > > > > Part 6: Thanks for clarifying Q4, and interesting discussion about Tables 1 and 2.
> > > > > > > > > >
> > > > > > > > > > Part 7: Again, interesting discussion and definitely something I suggest the authors include in the supplementary material.
> > > > > > > > > >
> > > > > > > > > > Part 8: What I meant to say is that NCA map inputs from and to the same domain, the authors are correct in pointing out that the domain need not be an image.
> > > > > > > > > > Thanks for the clarification regarding the baselines, I agree with the authors.
> > > > > > > > > > The texture generation results are probably worth a paper of their own, but they would definitely make a nice addition to the supplementary material.
> > > > > > > > > >
> > > > > > > > > > Part 9: Thanks for clarifying.
> > > > > > > > > >
> > > > > > > > > > ---
> > > > > > > > > >
> > > > > > > > > > While advising the authors to add as much as possible of this interesting discussion to the paper/supplementary material, I am happy to raise my score to a full accept and I will vouch for the paper if needed in the reviewer discussion (although there seems to be quite the consensus). Good job.

---

### Official Review · Reviewer_EYVF · 2022-07-11

**Rating:** 7
**Confidence:** 3
**Soundness:** 3 good
**Presentation:** 3 good
**Contribution:** 2 fair

**Summary:**

This paper proposes a new family of Neural Cellular Automata (NCA) based on visual Transformer for the first time. The main difference from the plain ViT is the use of a specially localized self-attention mechanism that is still globally organized, thus yielding a parameter-efficient model.  Quantitative and qualitative experiments show that the proposed ViTCA outperforms U-Net, UNetCA, and ViT on various benchmarks. The learned representations are also verified via linear probes evaluation.

**Questions:**

- Choosing ViT as a baseline to show the superiority of the proposed ViTCA makes sense. But why do you adopt U-Net, the CNN architecture, as the baseline? And why the proposed ViTCA succeeds while U-net fails? The authors do not describe the motivation and the reasons for such an experimental setup.
- Linear probes have been popular evaluation protocols to show the learning representation via self-supervised learning. I am wondering how does the proposed model perform using fine-tuning? Can the authors provide fine-tuning results or explain the reason why it is not good if fine-tuning performs poorly?
- The proposed model requires iterative training to converge. How can you ensure the stability of the proposed model?
- Performing experiments on MNIST and CIFAR 10 datasets is OK. It is better to validate the performance on larger datasets to show the practicality of the proposed model. The authors can either provide some analysis if the performance is not good.

**Limitations:**

The authors have discussed the limitations in Sec 5.

**Strengths And Weaknesses:**

Strengths:
+ This paper provides a new perspective on neural cellular automata with a visual Transformer architecture. Although it is a combination of problem and architecture, the authors propose an effective self-attention scheme to improve the model.
+ The experiments are relatively complete and can verify the proposed contributions.
+ The paper is well organized and provides informative figures.
Weaknesses:
- The biggest problem is that the authors do not explain why U-Net is adopted as a baseline model for comparison. This makes experimental comparisons confusing. Did you choose U-net as a CNN structure for comparison?
- It is better to validate the performance of linear probes on larger datasets since experiments are only evaluated on small MNIST and CIFAR 10 datasets. However, I do not mean that this will be the reason for the rejection of the paper.

---

> ### Author Response · Authors · 2022-08-02
> **Response to Reviewer EYVF (pt. 1)**
>
> We appreciate the extensive feedback and thank you for acknowledging the effectiveness of our proposed NCA model. Below we have provided our responses to your questions. We have also added a top-level comment to summarize the changes made in response to all four reviewers' comments and questions.
>
> **Q1**: _Choosing ViT as a baseline to show the superiority of the proposed ViTCA makes sense. But why do you adopt U-Net, the CNN architecture, as the baseline? And why the proposed ViTCA succeeds while U-net fails? The authors do not describe the motivation and the reasons for such an experimental setup._
>
> **Author response**: These are fair and valid questions. We believed the motivations for using a U-Net for comparison on denoising autoencoding performance were implied considering that U-Nets are widely used as "default" feedforward neural architectures that perform well on a variety of vision-based tasks, such as semantic segmentation, image classification, optical flow estimation, denoising autoencoding, etc. Before vision transformers, they were the dominant class of models for image classification. However, they remain highly competitive as backbone architectures for several vision-based tasks, especially image synthesis (of which denoising autoencoding is a subclass of). We cite [37] in the paper as an example of a competitive U-Net-based denoising autoencoder, of which we base our U-Net and UNetCA off of. We also point to [1] below which is a recent diffusion model which uses a U-Net backbone to provide state of the art results on text-conditioned image synthesis. To summarize, we tried to cover our evaluative bases by framing the comparisons as between two popular classes of image processing models: vision transformers (ViT) and convolutional neural nets (CNNs), and between two frameworks: NCA and non-NCA, of which the intersection of NCA and ViT (NCA + ViT = ViTCA) was what we were motivating.

---

> > ### Author Response · Authors · 2022-08-02
> > **Response to Reviewer EYVF (pt. 2)**
> >
> > **Q2**: _Linear probes have been popular evaluation protocols to show the learning representation via self-supervised learning. I am wondering how does the proposed model perform using fine-tuning? Can the authors provide fine-tuning results or explain the reason why it is not good if fine-tuning performs poorly?_
> >
> > **Author response**: This is an excellent question and we are glad that you asked, as we did in fact have fine-tuning results but unfortunately did not have the time to complete all experiments and to include what we had in the main manuscript or Appendix. However, we are happy to share them below, with some commentary. We only have fine-tuning results between the baselines and not the variants. Before sharing the three tables, we would like to explain the setup. The representations used were the same ones used for linear probing, the only differences being that the pre-trained models did not have their weights frozen and that the denoising autoencoding loss was still used. This required us to run the model twice per training iteration: one with a clean input for the classification loss as done with linear probing, and one with a noisy input for the denoising loss as done with our previous denoising training process. As you can infer, this was a part of the reason it was difficult for us to obtain results as there were issues with memory usage that weren't easily solvable with gradient checkpointing and issues with computation time.
> >
> >
> > |        | Acc. | # Params. |
> > |:------:|:----:|:---------:|
> > |  U-Net | 99.2 |   119.8K  |
> > |   ViT  | 94.6 |    1.4M   |
> > | UNetCA | 96.8 |   379.7K  |
> > |  ViTCA | 98.0 |   419.4K  |
> >
> > **Table 1:** Fine-tuning test accuracies of baseline models on MNIST. All models were pre-trained for denoising autoencoding then fine-tuned.
> >
> >
> > |        | Acc. | # Params. |
> > |:------:|:----:|:---------:|
> > |  U-Net | 92.0 |   119.8K  |
> > |   ViT  | 87.2 |    1.4M   |
> > | UNetCA | 10.0 |   379.7K  |
> > |  ViTCA | 89.6 |   419.4K  |
> >
> > **Table 2:** Fine-tuning test accuracies of baseline models on FashionMNIST. All models were pre-trained for denoising autoencoding then fine-tuned.
> >
> >
> > |        | Acc. | # Params. |
> > |:------:|:----:|:---------:|
> > |  U-Net | 73.4 |   122.0K  |
> > |   ViT  | 57.2 |    1.4M   |
> > | UNetCA | 65.5 |   381.7   |
> > |  ViTCA | 57.1 |   420.2K  |
> >
> > **Table 3:** Fine-tuning test accuracies of baseline models on CIFAR-10. All models were pre-trained for denoising autoencoding then fine-tuned.
> >
> > Note that the number of parameters listed in the above tables are the total number of trainable parameters, which in this case is the number of parameters for the pre-trained model plus the number of parameters for the linear classifier. This is in contrast to the number of parameters for linear probing where it's _only_ the number of parameters for the linear classifier since the pre-trained models are frozen.
> >
> > The above tables show some interesting results that motivate more experimentation for future work. The most obvious conclusion is that training a linear classifier on the U-Net's bottleneck features while permitting U-Net's weights to be fine-tuned results in the best classification performance. Despite this, classifying using ViTCA's cell hidden information results in better performance than with UNetCA's and and ViT's respective representations on MNIST and FashionMNIST. Oddly enough, classifying with ViTCA's cell hidden information is marginally worse than with ViT's representation on CIFAR-10 and much worse than with UNetCA's cell hidden representation. We do not have any immediate explanation as to why this happens and it is difficult to arrive at any definitive conclusion without further testing and analysis.

---

> > > ### Author Response · Authors · 2022-08-02
> > > **Response to Reviewer EYVF (pt. 3)**
> > >
> > > **Q3**: _The proposed model requires iterative training to converge. How can you ensure the stability of the proposed model?_
> > >
> > > **Author response**:
> > > We are unsure whether you are asking about the stability of the training process or the stability of cell states after they have converged to a solution during inference. We will attempt to answer both:
> > >
> > > _Ensuring training stability (preventing loss and gradient-related training issues):_
> > >
> > > There are a couple of tricks we implement to keep training stable and manageable. First, during training we zero initialize the weights and biases of the final layer of our CA-based models (mentioned in L199) which allows for the update rule to exhibit a "do nothing" initial behaviour (this idea was taken from [30] as referenced in the paper). Since cell output channels are initialized to 0.5, cell hidden channels are initialized to 0, target image RGB values are in the range [0, 1], and overflow loss occurs when the output RGB values are past the range [0, 1] and when the hidden values are past the range [-1, 1], this results in the first CA training iteration having a loss with low magnitude. This means gradients are also of low magnitude and the second training iteration will not result in a dramatic change in the loss. This makes the training process more stable. We experimented with initializing the weights and biases of the final layer the same as with other layers (He initialization) and noticed the training process quickly grow unstable, which was most unstable with UNetCA since it lacked any normalization layers like ViTCA does. Second, ViTCA's layer normalizations, which helps keep internal activations within a manageable and stable range, which in turn improves training stability. Third, before applying gradients accumulated after the $T$ recurrent cell updates, we normalize them. Line 19 in Algorithm 1 (in Appendix A) shows this.
> > >
> > > _Ensuring cell state stability after convergence during inference:_
> > >
> > > The overflow loss and pool sampling training scheme are the two techniques used to directly encourage long-term cell state stability during inference time, _i.e.,_ to have a cell update rule that does not cause the cells' values to diverge after they have settled on a stable point. The overflow loss encourages the update rule to keep cell states within a certain range of values. The pool sampling training strategy of sampling from a pool of cell grids which were previously iterated upon and iterating upon them again before putting them back into the pool is a way of training the cell update rule (ViTCA in this case, but it applies to all NCAs since this is a common NCA training method as mentioned in the paper) to always strive to keep cells at the correct solution (the point that results in the lowest loss). As can be inferred by the max size of the pool and that at each _odd_ training iteration it grows with a minibatch of new cell grids, there are multiple points during training when the sampled cell grid (_i.e.,_ during _even_ train iterations) has already reached its converged point (for example, at its 1000th iteration) yet ViTCA is still required to update it and maintain a low loss. We point to the Growing CA Distill article [30] (in the paper's references) for a well-presented explanation of the pool sampling process they originated. We also point to the cart-pole NCA paper [22] (in the paper's references) for more on the overflow loss.

---

> > > > ### Author Response · Authors · 2022-08-02
> > > > **Response to Reviewer EYVF (pt. 4)**
> > > >
> > > > **Q4**: _Performing experiments on MNIST and CIFAR 10 datasets is OK. It is better to validate the performance on larger datasets to show the practicality of the proposed model. The authors can either provide some analysis if the performance is not good._
> > > >
> > > > **Author response**: We are not sure if this is a question so we are assuming it is a critique on the size of the datasets that were tested on for linear probing and will answer accodrdingly. To succicintly answer your concern, we agree that it would be useful to validate linear probe performance on datasets larger than MNIST, FashionMNIST, and CIFAR-10. However, considering that the linear probe experiments were meant as a brief look into the learned representations of the compared models and not as a primary motivation for ViTCA, when we triaged our list of experiments we had to forego probing with models trained on larger datasets in the interest of time. We decided not to use our Celeb-A pre-trained models since Celeb-A has 40 annotations per image, which would result in a difficult-to-parse PCA visualization. MNIST, FashionMNIST, and CIFAR-10 each have 10 classes, making for an easier qualitative verification of the linear separation of the learned representations when visualizing their PCA space. We decided not to use our Tiny ImageNet pre-trained models because of unforeseen memory limitations, which made it incredibly time-consuming to work around it with gradient checkpointing (since it reduces memory consumption at the cost of computation time). This is unlike the memory limitations found when training ViTCA and UNetCA. Specifically, since the method of linear probing was a linear layer on the concatenation of _all_ cells, and the cell grid spatial size being $64 \times 64$ instead of $32 \times 32$ because of the Tiny ImageNet image size, memory usage was too time-consuming to manage in time for submission. On top of that, we had the same concern with the number of classes as we did with Celeb-A since Tiny ImageNet has 200 classes. All this being considered, we settled on the three datasets mentioned in the paper. To conclude, this could be an interesting avenue to explore in the future for a paper-specific analysis on the learned representations of NCA models in general, which [21] also encourages.
> > > >
> > > > **References**
> > > >
> > > > [1] A. Ramesh, P. Dhariwal, A. Nichol, C. Chu, and M. Chen, "Hierarchical text-conditional image generation with CLIP latents," _arXiv preprint arXiv:2204.06125_, 2022.

---

### Official Review · Reviewer_GAh5 · 2022-07-12

**Rating:** 7
**Confidence:** 4
**Soundness:** 4 excellent
**Presentation:** 3 good
**Contribution:** 3 good

**Summary:**

This paper proposes a novel combination of vision transformers and Neural Cellular Automata (NCAs), and uses them to create denoising autoencoders. An NCA is essentially a cellular automata with the node updates being performed by a neural net. The ViTCA proposed here adds to this attention heads that are only focused on neighboring cells, and includes positional encodings. The paper demonstrates superior performance to a ViT on denoising autoencoder tasks, and demonstrates the robustness of the model to various perturbations.

I have read the authors' responses and am happy to update my rating to a 7. Their responses were quite thorough, although (correct me if I'm wrong) they didn't respond to the "one task" critique.




**Questions:**

1. What are the essential differences between your model and a universal transformer with local attention? Universal transformers simply use the same weights over and over, so they are recurrent (Dehghani, et al. 2018). One difference is the asynchronous update rule. Others?

2. On lines 209-210, you mention that ViTCA outperforms the other baselines on 10 out of the 18 datasets, although you are reporting only 6. Are there 12 more datasets you aren’t telling us about??

3. Instead of overflow loss, have you tried just using nonlinear activation functions that stay within the desired range?

Mostafa Dehghani, Stephan Gouws, Oriol Vinyals, Jakob Uszkoreit, Łukasz Kaiser (2018) Universal Transformers. arXiv:1807.03819.

**Limitations:**

Yes.

**Strengths And Weaknesses:**

Strengths

1. This is a novel combination of transformers and neural cellular automata.

2. The results look very good compared to a vanilla ViT. E.g., Figure 1 and parts of Figure 2 make a good case for the model.

3. While the baseline ViT used for comparison has slightly fewer parameters than the baseline ViTCA, making for an unfair comparison, smaller versions of the ViTCA (e.g., ViTCA-i) also outperform it with many fewer parameters. This could be better emphasized in the paper. Although this does beg the question - how would an inverted bottleneck ViT with a similar number of parameters perform on this task?

4. The writing is fairly clear, although Figures 2 and 3 are not.

5. I believe, given the results, that this work is significant, but they only demonstrated efficacy on one task, which reduces enthusiasm somewhat.


Weaknesses:

1. It is very hard to understand what is going on in Figures 2 and 3. I can make some sense of Figure 3, but Figure 2 is a mystery. I don’t know what a “splat map” is, and I can’t tell that the cells are focusing on foreground, noise and background, as mentioned in the caption. The authors should try showing these figures to a colleague and finding out what they have to explain to get the idea across, and then simplify the figure to make this clear.

2. This reviewer, at least, would have appreciated a demonstration of this architecture on another problem (e.g., semantic segmentation), rather than a deep dive into the model robustness to ablations and various perturbations (section 4.1.1-4.1.3), which could have been relegated to the supplementary material.

3. This model is similar to a universal transformer, which also uses essentially depth 1 (since the parameters are used over and over), with iterative processing.

---

> ### Author Response · Authors · 2022-08-02
> **Response to Reviewer GAh5 (pt. 1)**
>
> Thank you for your constructive feedback and for recognizing the novelty and significance of our work. Below we have provided our responses to your questions as well as to some of your comments on the strengths and weaknesses of the paper. We have also added a top-level comment to summarize the changes made in response to all four reviewers' comments and questions.
>
> **Strength 3**: _While the baseline ViT used for comparison has slightly fewer parameters than the baseline ViTCA, making for an unfair comparison, smaller versions of the ViTCA (e.g., ViTCA-i) also outperform it with many fewer parameters. This could be better emphasized in the paper. Although this does beg the question - how would an inverted bottleneck ViT with a similar number of parameters perform on this task?_
>
> **Author response**: You raise an excellent question on how an inverted bottleneck ViT would perform. Fortunately, we had performed this experiment prior to submission. Unfortunately, we did not include it due to space and time constraints. Before we provide the table of results and commentary we would like to respond to your comment on the fairness of the comparison between the baseline ViT and baseline ViTCA on the grounds of slightly differing parameter counts.
>
> We feel that ViT having slightly fewer parameters than ViTCA should not disqualify the fairness of comparisons between the two. ViTCA and ViT use the same backbone architecture and, parameter-wise, only differ in the size of their respective inputs and outputs, which affects their embedding layer and head layer parameter counts (keep in mind that the attention size does not affect parameter count). Their internal representation dimensionality remain in parity. ViTCA operates with higher dimensional inputs, _i.e.,_ cells, and higher dimensional outputs, _i.e.,_ cell updates. Each cell contains an input image patch (for our experiments, input/output image patches are $1 \times 1$, meaning a single RGB pixel), an output image patch (produced from the previous iteration), hidden information, and (optionally) positional information (which would allow foregoing the positional encoding used within the transformer). Cell updates consist of updating the cell output channels and the cell hidden channels. ViT, on the other hand, consumes only input image patches and produces only output image patches. If we wanted parameter parity between ViT and ViTCA, we would have to change the inputs/outputs of ViT to match that of ViTCA, specifically, that the inputs are cells and the outputs are cell updates. The issue with this approach is that without recurrent updates, the cell-based setup would be pointless since the cell hidden information isn't being updated and previous cell states are not being considered. Furthermore, without recurrent updates, the additional information in the input, namely the cell hidden information and cell output information (which are initialized to 0 and 0.5, respectively) will act as useless noise for ViT. Once you add recurrent updates, then it is only one step away from being ViTCA: localizing attention. The main motivation of our paper is that by moving ViT to a CA framework, without changing its internal representation dimensionality and localizing its attention (which also doesn't change dimensionality), you end up with a more parameter efficient ViT.

---

> > ### Author Response · Authors · 2022-08-02
> > **Response to Reviewer GAh5 (pt. 2)**
> >
> > Here we provide four tables of denoising autoencoding results on the test sets for LandCoverRep, CelebA, MNIST, and FashionMNIST. We did not have results on CIFAR-10 or TinyImageNet due to time constraints.
> >
> > |         |  PSNR |  SSIM | LPIPS | # Params. |
> > |:-------:|:-----:|:-----:|:-----:|:---------:|
> > | VITCA-i | **33.49** | **0.929** | **0.108** |   54.7K   |
> > | ViT-i   | 29.99 | 0.878 | 0.154 |   50.4K   |
> >
> > **Table 1**: Comparing denoising autoencoding test results on LandCoverRep between inverted bottleneck variants of ViT and ViTCA.
> >
> >
> > |         |  PSNR |  SSIM | LPIPS | # Params. |
> > |:-------:|:-----:|:-----:|:-----:|:---------:|
> > | VITCA-i | **26.10** | **0.904** | **0.074** |   54.7K   |
> > | ViT-i   | 18.87 | 0.737 | 0.310 |   50.4K   |
> >
> > **Table 2**: Comparing denoising autoencoding test results on CelebA between inverted bottleneck variants of ViT and ViTCA.
> >
> >
> > |         |    PSNR   |    SSIM   |   LPIPS   | # Params. |
> > |:-------:|:---------:|:---------:|:---------:|:---------:|
> > | VITCA-i | **26.03** | **0.930** | **0.033** |   54.3K   |
> > | ViT-i   | 14.76     | 0.537     | 0.313     |   50.1K   |
> >
> > **Table 3**: Comparing denoising autoencoding test results on MNIST between inverted bottleneck variants of ViT and ViTCA.
> >
> >
> > |         |    PSNR   |    SSIM   |   LPIPS   | # Params. |
> > |:-------:|:---------:|:---------:|:---------:|:---------:|
> > | VITCA-i | **22.84** | **0.827** | **0.139** |   54.3K   |
> > | ViT-i   | 15.64     | 0.502     | 0.405     |   50.1K   |
> >
> > **Table 4**: Comparing denoising autoencoding test results on FashionMNIST between inverted bottleneck variants of ViT and ViTCA.
> >
> > Although ViTCA-i has **~8%** more parameters than ViT-i (due to differences in input/output dimensionality), when considering that the scale of PSNR is logarithmic, that the range of SSIM is [0, 1], and that the range of LPIPS is [0, 1], there is a substantial jump in performance in ViTCA-i compared to ViT-i. On average, there is a **43.1%** relative improvement in PSNR, a **23.4%** absolute improvement in SSIM, and a **20.7%** absolute improvement in LPIPS.

---

> > > ### Author Response · Authors · 2022-08-02
> > > **Response to Reviewer GAh5 (pt. 3)**
> > >
> > > **Q1**: _What are the essential differences between your model and a universal transformer with local attention? Universal transformers simply use the same weights over and over, so they are recurrent (Dehghani, et al. 2018). One difference is the asynchronous update rule. Others?_
> > >
> > > **Author response**:
> > > You raise a completely valid concern and listed below are other essential differences between ViTCA and Universal Transformers (UT):
> > > 1. In each time step, UT's attention and self-attention computations are computed across all tokens, _i.e.,_ _globally_. In each time step, ViTCA's self-attention is computed across a localized neighbourhood about each cell (cells are treated as tokens), circumventing the quadratic computational complexity of global self-attention while still maintaining global information propagation through recurrent cell updates.
> > > 2. UT is designed for natural language processing, ViTCA is designed for image processing, resulting in different computational pipelines. UT first embeds each symbol (UT treats symbols as tokens, examples of symbols are words and phonemes) in the input sequence then uses its recurrent encoder to recurrently update the embedding with self-attention and its transition function. After $T$ steps, the embedding is then fed to the recurrent decoder where it uses attention on the encoder's embedding, self-attention on its own embedding, and its own transition function to update its own embedding. The decoder's embedding comes from embedding its current sequence of output symbols. The decoder produces the next output symbol at each decoder iteration with an argmax on its output probabilities which are computed using a softmax on the output of its transition function after $T$ steps.
> > > Basically, UT operates in two stages in sequence: an encoder stage followed by a decoder stage, each with their own transformer where the encoder has one attention block and the decoder has _two_ to attend to both its own tokens and the encoder's. After the encoder iterates $T$ times, the decoder iterates $T$ times to produce *one* output token. **This means the decoder needs to iterate $TN$ times to produce $N$ output tokens**. It's an autoregressive model. ViTCA, on the other hand has its encoder and decoder operate in lockstep over cell update iterations. The encoder processes tokens which are then fed to an MLP head acting as its decoder to produce cell updates for all cells, all in one time step, done asynchronously, using a single attention block.
> > > 3. Recurrent updates in ViTCA are performed on tokens (_i.e.,_ cells) while recurrent updates in UT are performed on token _embeddings_ (_i.e.,_ symbol embeddings). In other words, ViTCA uses cells as its state representation and UT uses symbol embeddings as its state representation. At the start of each iteration, ViTCA feeds cells to an embedding layer, while UT does not. Another important difference between the token representations used in the two models is that cells not only contain the input information (like an image patch or even a symbol if need be) but also contain hidden information that can be used to facilitate inter-cell communication, output information for storing the output from the previous iteration, and optionally, positional information. UT's encoder and decoder tokens do not contain hidden information, they do not contain the previous iteration's output, and they're not designed to include additional information like positional encodings. Also, in ViTCA, the output is directly extracted from tokens (cells) while in UT the output is autoregressively decoded from a decoder transformer operating on encoder token embeddings and decoder token embeddings.

---

> > > > ### Author Response · Authors · 2022-08-02
> > > > **Response to Reviewer GAh5 (pt. 4)**
> > > >
> > > > **Q2**: _On lines 209-210, you mention that ViTCA outperforms the other baselines on 10 out of the 18 datasets, although you are reporting only 6. Are there 12 more datasets you aren’t telling us about??_
> > > >
> > > > **Author response**: The sentence in question: "Amongst baselines, ViTCA outperforms on most metrics across the majority of datasets used (10 out of 18)."
> > > >
> > > > To answer your question, no, there are not any more datasets beyond the six described in the paper. To clarify the meaning of the sentence, it is referring to 10 out of the total number of measurements made across all datasets, of which there are 18 measurements. Specifically, there are 6 datasets, there are 3 metrics used (PSNR, SSIM, LPIPS), so there are 6 * 3 = 18 measurements in total. ViTCA gets the best result on 10 of those measurements when comparing to the other baseline models in Table 1.

---

> > > > > ### Author Response · Authors · 2022-08-02
> > > > > **Response to Reviewer GAh5 (pt. 5)**
> > > > >
> > > > > **Q3**: _Instead of overflow loss, have you tried just using nonlinear activation functions that stay within the desired range?_
> > > > >
> > > > > **Author response**: We believe there is a misunderstanding of the purpose of the overflow loss. We are assuming that you are referring to restricting the cell update range with a non-linear activation, please correct us if we are assuming incorrectly.
> > > > >
> > > > > The overflow loss is not meant for restricting the range of cell update values, but rather for encouraging the range of cell output values to be within [0, 1] and cell hidden values to be within [-1, 1]. It is intended to encourage long term cell state stability, inspired by [20] and [22] (in the paper's references). On the other hand, the cell _updates_ produced by the update rule, which _update cell values_, are unbounded. This allows for additive and subtractive cell updates, as done by [20], [21], and [30]. Despite this, we _did_ experiment with restricting the bounds of cell updates by using a sigmoid, tanh, or ReLU activation after the MLP head's linear layer (which produces the update each iteration).
> > > > >
> > > > > Before explaining our results for each, it's important to highlight the tricks we implemented to keep training stable and manageable:
> > > > >
> > > > > First, during training we zero initialize the weights and biases of the final layer of our CA-based models (mentioned in L199) which allows for the update rule to exhibit a "do nothing" initial behaviour (this idea was taken from [30]). Since cell output channels are initialized to 0.5, cell hidden channels are initialized to 0, target image RGB values are in the range [0, 1], and overflow loss occurs when the output RGB values are past the range [0, 1] and when the hidden values are past the range [-1, 1], this results in the first CA training iteration having a loss with low magnitude. This means gradients are also of low magnitude and the second training iteration will not result in a dramatic change in the loss. This makes the training process more stable. We experimented with initializing the weights and biases of the final layer the same as with other layers (He initialization) and noticed the training process quickly grow unstable, which was most unstable with UNetCA since it lacked any normalization layers like ViTCA does.
> > > > >
> > > > > Second, ViTCA's layer normalizations, which helps keep internal activations within a manageable and stable range, which in turn improves training stability.
> > > > >
> > > > > Third, before applying gradients accumulated after the $T$ recurrent cell updates, we normalize them. Line 19 in Algorithm 1 (in Appendix A) shows this.
> > > > >
> > > > > Now we can get into the results of our activation experiments:
> > > > >
> > > > > _Sigmoid:_
> > > > > This eliminated the "do nothing" initial behaviour since it transformed the zero output of the final layer (since its weights and biases were initialized to 0) to a 0.5 output, which resulted in unstable training by immediately increasing cell values. Furthermore, since a sigmoid outputs values between (0, 1), this meant cell updates were always additive and were never subtractive, which caused explosive cell state growth and unstable training. On a related note, recurrent models are much more susceptible to the exploding gradient problem when using sigmoid activations. This typically leads to using ReLUs instead.
> > > > >
> > > > > _ReLU:_
> > > > > As experienced by [30], using a ReLU resulted in only positive cell updates and no subtractive cell updates. Also, since it's unbounded, it quickly resulted in explosive cell state growth, which destabilized training.
> > > > >
> > > > > _Tanh:_
> > > > > Tanh allowed for both additive and subtractive cell state updates. Although its derivative quickly shrinks with non-zero inputs (even with input values with an absolute value < 1)---which can cause vanishing gradients---our use of gradient normalization, our weight/bias initialization (He init except for final layer that uses zero init), the fact that target RGB values are in the range [0, 1], and initial cell output values being 0.5 meant that vanishing gradients wouldn't be much of a problem, at least during the start of training when inputs to tanh were small. The problem with using tanh was that it was still susceptible to vanishing gradients over time, despite our mitigations, and it slowed down training progress since it would prevent being able to quickly update cells to the desired ranges (which would also result in worse training losses). We found it preferable to just not use an activation function since the training process was already quite stable.

---

### Official Review · Reviewer_gcLv · 2022-07-13

**Rating:** 8
**Confidence:** 3
**Soundness:** 4 excellent
**Presentation:** 4 excellent
**Contribution:** 4 excellent

**Summary:**

The authors combine the Vision Transformer architecture with Neural Cellula Automata (NCA, which combines artificial neural networks with cellula automata). They demonstrate that introducing self-attention and positional encodings to NCAs significantly improves performance. One of the main contributions of this work is the authors adaptation of to the global self-attention mechanism to respect the localisation condition for NCAs. The receptive field of individual cells is implicitly grown due to cell updates during CA iterations such that there is a global propagation of information from localised self-attention. This self-organisation is shown to give performance advantages when compared to other architectures.




**Questions:**

One of the main contributions here is the propagation of local information through CA. However, with large CA systems, I can imagine that there is a separation of timescales between local information propagation and the information being implicitly available to make global changes. Have the authors considered (empirically or theoretically) the rate of information propagation through the system and whether this scales with system size?

**Limitations:**

As the authors note, there are memory constraints with using recurrent models such as CA, which they offer potential future solutions to.

**Strengths And Weaknesses:**

This is a welcome extension of NCAs to include self-attention and, as such, is highly original. As I understand it, one of the main challenges of achieving this has been how to combine the local computation of NCAs with global updates in ViT, without having highly inefficient learning process. The main contribution from this paper has been an attempt to do exactly this.

The authors have provided a thorough analysis into the ViTCA, across six benchmark datasets and multiple baseline architectures including ViT, U-Nets and UNetCA. They find that ViTCA can outperform other approaches across on most benchmark datasets. Particularly interesting is the discussion of cell states, as compared to UNetCA, where the authors find that ViTCA is maintains cell stability even when damaged whereas UNetCA diverges.

In general, I found the manuscript clear and full of information. Some of the figures had almost too much information and it was difficult to relate the caption to the figure. For example, in Figure 2 it took me some time to understand what ‘middle' and 'left,right' exactly referred to. Overall, I thought it was well structured and presented.

---

> ### Author Response · Authors · 2022-08-02
> **Response to Reviewer gcLv**
>
> We are very thankful for your positive feedback and recognition of the contributions of ViTCA. Please find our response to your questions and comments below. We have also added a top-level comment to summarize the changes made in response to all four reviewers' comments and questions.
>
> **Q1**: _One of the main contributions here is the propagation of local information through CA. However, with large CA systems, I can imagine that there is a separation of timescales between local information propagation and the information being implicitly available to make global changes. Have the authors considered (empirically or theoretically) the rate of information propagation through the system and whether this scales with system size?_
>
> **Author response**: This is a great question. Yes, this was considered and discussed at some point during the project, with rudimentary calculations made to estimate the rate of information propagation over time. Excluding the effects of asynchronous updating, the rate of information propagation due to the recurrent applications of localized self-attention (at a $3 \times 3$ attention neighbourhood size) is analogous to how the theoretical receptive field of neurons, with respect to the input, linearly grows through layers of $3 \times 3$ convolutions. So the same formulas apply. Specifically, when looking at just the height dimension (same applies for width),
>
> $$r_T = \left(\sum_{t=1}^{T} \left( N_H - 1 \right) \right) + 1 \quad ,$$
>
> where $r_T$ denotes the height-specific receptive field of the final attention operation after $T$ iterations and $N_H$ is the attention height. So for example, with ViTCA's $3 \times 3$ attention, after one iteration each cell is updated using information from their $3 \times 3$ neighbourhood. After two iterations each cell is updated using information, implicitly, from their $5 \times 5$ neighbourhood. Third iteration, $7 \times 7$ neighbourhood, and so on. Their receptive fields grow linearly with respect to time and you can interpret this in a transposed manner to say that the rate of spatial information propagation from one cell is linear in the exact same way (_i.e.,_ it's the same equation). We suggest looking at [1] (references provided below) for more commentary on receptive fields and for interactive demos visualizing receptive field growth, which similarly applies to the rate of information propagation from localized attention.
>
> The rate of information propagation does not scale with the size of the cell grid (assuming that is what you meant by system size), it scales with the size of the cell neighbourhood (for ViTCA, the size of its attention neighbourhood and for UNetCA, the size of its first convolution). Although in this case the scale is linear, one can increasingly dilate or grow the attention size over time for an exponential growth in the rate of information propagation, although we feel this pushes against the philosophy of local computations in cellular automata (although locality is relative, after all). As a matter of fact, considering that the attention size does not affect the number of parameters, an experiment we would like to investigate in future work is to allow ViTCA to dynamically alter the attention neighbourhood size or dilation to maximize rate of information propagation, maximize accuracy, minimize unnecessary compute (if attention past a certain neighbourhood size isn't useful, shrink it before the next iteration), and staying within some user-chosen max compute budget. Another one of our experiment ideas for future work is to visualize the actual region of influence of a cell over time, similar to Figure 14 in the cart-pole NCA paper [2].
>
> **References**
>
> [1] A. Araujo, W. Norris, and J. Sim, "Computing receptive fields of convolutional neural networks," _Distill_, 2019, https://distill.pub/2019/computing-receptive-fields/ .
>
> [2] A. Variengien, S. Nichele, T. Glover, and S. Pontes-Filho, "Towards self-organized control: Using neural cellular automata to robustly control a cart-pole agent," in _Innovations in Machine Intelligence (IMI)_, 2021, pp. 1-14.

---

### Author Response · Authors · 2022-08-02
**Overall Author Response to Reviewers (pt. 1)**

We are grateful to all the reviewers for their constructive feedback and appreciate that they all recognize the novelty of enveloping Vision Transformers within an NCA framework. A quick note before proceeding to read our responses to your respective comments:
1. All line number references are for the _unrevised_ manuscript, not the revised one we have submitted.
2. Paper citation numbers refer to citation numbers listed in our unrevised manuscript unless otherwise stated since we provide an additional list of references at the final part of our multi-part responses.

Some common strengths listed amongst reviewers were:
- **S1**: The idea is novel.
- **S2**: The paper is clear, well-structured, and well-presented.
- **S3**: The experimentation is in-depth and complete.

Some common weaknesses listed amongst reviewers were:
- **W1**: Reviewers gcLv and GAh5 found some of the figures hard to follow. Figures 2 and 3 were specifically listed.

Common questions (with our answer) shared amongst reviewers were:
- **Q1**: _(from reviewers GAh5 and mu6v)_ Why weren't non-linear activations used to restrict the range of cell updates instead of an overflow loss?
- **Shortened author response**: _(please see our individual responses to each reviewer for more)_ We believe there is a misunderstanding of the purpose of the overflow loss. The overflow loss is not meant for restricting the range of cell update values, but rather for encouraging the range of cell _output_ values to be within [0, 1] and cell _hidden_ values to be within [-1, 1]. Overflow losses were used in [20] and [22] (in the paper's references) and are intended to encourage long term cell state stability. On the other hand, the cell _updates_ produced by the update rule, which _update cell values_, are unbounded. This allows for additive and subtractive cell updates, as done by [20], [21], and [30] (cited in the paper). Despite this, we _did_ experiment with restricting the bounds of cell updates by using a sigmoid, tanh, or ReLU activation after the MLP head's linear layer (which produces the update each iteration). In short, sigmoid caused explosive cell state growth, despite having an overflow loss, since its outputs were always positive; ReLU had the same effect; tanh made the training process prone to vanishing gradients and since it forced the update range to be between -1 and 1, it prevented quick corrective cell updates when cell state values were a bit too far past the overflow ranges, which in turn harmed training.

Modifications made to our manuscript in response to reviewer comments and questions:
- **M1**: Reviewer mu6v pointed out an inconsistency in the number of parameters listed in Table 3 between the 2-layer MLP with 100 hidden units and the one with 1000 hidden units, under the CIFAR10 column. The 2-layer MLP with 1000 hidden units should have 3.1M parameters and not 308.3K. We have corrected this mistake in the revised manuscript.
- **M2**: In light of some of reviewer mu6v's misunderstandings of the parameter counts in the tables, we have clarified the meaning of the parameter counts for the linear probe experiments in the caption of Table 3. The following sentence was added: "Parameter counts exclude fixed parameters." As such, they refer to only the parameters of the additional classifiers.  Although we would like to provide even further clarification, any additional words will push the manuscript past the 9 page limit. Further clarification is provided below in **C2**.
- **M3**: L244 shows "(1b) xy..." which should be "(1c) xy..." since (1b) already refers to the "sincos5xy" method of positional encoding. This has been fixed in the revised manuscript.

---

> ### Author Response · Authors · 2022-08-02
> **Overall Author Response to Reviewers (pt. 2)**
>
>
> Additional clarifications we would like to make in response to some questions and comments:
> - **C1**: As more heads are added in multi-head self-attention (MHSA), for example, going from $h = 2$ to $h = 3$, the number of parameters remain the same since the dimensionality of each head is equal to the embedding size $d$ divided by the number of heads $h$. This is typical and followed by the original ViT paper. We also show this in L130 in our (unrevised) manuscript. However, when going from single-head self-attention to multi-head, $h = 1$ to $h = 2$, the number of parameters _slightly increases_ since in MHSA the heads need to be consolidated together using an additional weight matrix before adding the result to the token embedding. This matrix is $\mathbf{W}$ in L132, which is not used when $h = 1$. Despite the performance gains in increasing the number of heads, as shown with ViTCA-32 in Table 2, it's important to note that it does negatively affect runtime performance with respect to compute time and memory usage (implied in L272-273), something we hope to alleviate in the future with custom CUDA kernels for the localized self-attention mechanism.
> - **C2**: To further clarify the meaning of parameter counts listed in the tables, they represent the number of _trainable_ parameters in the listed model for the respective experiment. Since Table 1 and 2 report denoising autoencoding test results, the parameter counts refer to the number of parameters in the listed models (since they're the ones being trained). In Table 3, since the experiment is for linear probing, the number of parameters are for the linear probe/classifier that is being trained on the features of the pre-trained fixed models. The last three models listed in Table 3 do not use pre-trained models since they're being trained directly on the input images, so the parameter counts listed are _their_ number of trainable parameters.
> - **C3**: We would like the reviewers to consider the relative and absolute scales of the PSNR, SSIM, and LPIPS metrics used and how that affects the interpretation of percentage of performance difference between the models. As a reminder, PSNR is on a logarthmic scale and SSIM and LPIPS are in a bounded range of [0, 1].
> - **C4**: The localized self-attention neighbourhood size does not affect the number of parameters but it does affect compute time and memory usage. Furthermore, by how localized self-attention is constructed, one can notice that it can be dynamically changed during runtime, which opens avenues for future work where ViTCA is trained to dynamically alter the attention neighbourhood size or dilation to maximize rate of information propagation, maximize accuracy, minimize unnecessary compute (if attention past a certain neighbourhood size isn't useful, shrink it before the next iteration), all while staying within some user-chosen max compute budget.
>
> Miscellaneous:
> - Reviewer mu6v cited texture synthesis as a more interesting task and ironically it was the first task we had tested on while working on this ViTCA project, since it stemmed from our interest in the Self-Organizing Textures work of [21]. We did not pursue this direction because of the following limitations. First, we felt it was too specialized of a task to infer meaningful conclusions on the inductive biases of ViTCA for generalized image synthesis-based tasks. Second, there is no training involved, only optimization on a single texture exemplar. Third, texture does not cover higher-level abstract representations and semantics which images in datasets like CelebA, CIFAR10, and Tiny ImageNet do. Finally, our other experimentation already fills the allowed space for our submission. However, we did intentionally use the LandCoverRep data to serve as a texture oriented evaluation since it can be viewed as essentially images of textures. Despite all this, we are delighted to share some results we had from these early experiments, linked in this [imgur link](https://imgur.com/a/5JwVHzW). To summarize what we observed from these experiments: compared to Texture ViTCA, Texture NCA takes a much longer time to output a texture which looks qualitatively worse, all while using 44% more parameters than Texture ViTCA. We hope to investigate this direction in the future, especially considering the exciting advancements made in [20] with its Optimal Transport Texture loss.

---

### Meta-Review · Area_Chair_3PKS · 2022-08-26

**Recommendation:** Accept
**Confidence:** Certain

**Metareview:**

As summarized by reviewer GAh5, this paper proposes a novel combination of vision transformers and Neural Cellular Automata (NCAs), and uses them to create denoising autoencoders. An NCA is essentially a cellular automaton with the node updates being performed by a neural net. The ViTCA proposed here adds to this attention heads that are only focused on neighboring cells, and includes positional encodings. The paper demonstrates superior performance to a ViT on denoising autoencoder tasks, and demonstrates the robustness of the model to various perturbations.

All reviewers agree that this is a novel contribution to NCAs. By replacing convolutions with vision transformers, it opens up many possibilities (such as scale). The experiments were conducted with detail and clarity. I do agree with reviewer mu6v, that while the quality of the paper is good, the experiments go fairly in-depth although on just one task of image denoising (which is less interesting than texture generation or other forms of morphogenesis), but even in its current form, it definitely meets the bar for a solid acceptance recommendation from me.


**Award:**

No

---

### Decision · Program_Chairs · 2022-09-14

Accept